# Near-Optimal-Sample Estimators for Spherical Gaussian Mixtures

**Jayadev Acharya**[*]
MIT
jayadev@mit.edu

**Ashkan Jafarpour, Alon Orlitsky, Ananda Theertha Suresh**
UC San Diego
{ashkan, alon, asuresh}@ucsd.edu

## Abstract

Many important distributions are high dimensional, and often they can be modeled as Gaussian mixtures. We derive the first sample-efficient polynomial-time estimator for high-dimensional spherical Gaussian mixtures. Based on intuitive spectral reasoning, it approximates mixtures of $k$ spherical Gaussians in $d$-dimensions to within $\ell_1$ distance $\epsilon$ using $\mathcal{O}(dk^9(\log^2 d)/\epsilon^4)$ samples and $\mathcal{O}_{k,\epsilon}(d^3 \log^5 d)$ computation time. Conversely, we show that any estimator requires $\Omega(dk/\epsilon^2)$ samples, hence the algorithm's sample complexity is nearly optimal in the dimension. The implied time-complexity factor $\mathcal{O}_{k,\epsilon}$ is exponential in $k$, but much smaller than previously known.

We also construct a simple estimator for one-dimensional Gaussian mixtures that uses $\widetilde{\mathcal{O}}(k/\epsilon^2)$ samples and $\widetilde{\mathcal{O}}((k/\epsilon)^{3k+1})$ computation time.

## 1 Introduction

### 1.1 Background

Meaningful information often resides in high-dimensional spaces: voice signals are expressed in many frequency bands, credit ratings are influenced by multiple parameters, and document topics are manifested in the prevalence of numerous words. Some applications, such as topic modeling and genomic analysis consider data in over 1000 dimensions [31, 14]. Typically, information can be generated by different types of sources: voice is spoken by men or women, credit parameters correspond to wealthy or poor individuals, and documents address topics such as sports or politics. In such cases the overall data follow a mixture distribution [26, 27]. Mixtures of high-dimensional distributions are therefore central to the understanding and processing of many natural phenomena. Methods for recovering the mixture components from the data have consequently been extensively studied by statisticians, engineers, and computer scientists.

Initially, heuristic methods such as expectation-maximization were developed [25, 21]. Over the past decade, rigorous algorithms were derived to recover mixtures of $d$-dimensional spherical Gaussians [10, 18, 4, 8, 29] and general Gaussians [9, 2, 5, 19, 22, 3]. Many of these algorithms consider mixtures where the $\ell_1$ distance between the mixture components is $2 - o_d(1)$, namely approaches the maximum of 2 as $d$ increases. They identify the distribution components in time and samples that grow polynomially in $d$. Recently, [5, 19, 22] showed that the parameters of any $k$-component $d$-dimensional Gaussian mixture can be recovered in time and samples that grow as a high-degree polynomial in $d$ and exponentially in $k$.

A different approach that avoids the large component-distance requirement and the high time and sample complexity, considers a slightly relaxed notion of approximation, sometimes called *PAC learning* [20], or *proper learning*, that does not approximate each mixture component, but instead

---

[*]Author was a student at UC San Diego at the time of this work

derives a mixture distribution that is close to the original one. Specifically, given a distance bound $\epsilon > 0$, error probability $\delta > 0$, and samples from the underlying mixture $\mathbf{f}$, where we use boldface letters for $d$-dimensional objects, PAC learning seeks a mixture estimate $\hat{\mathbf{f}}$ with at most $k$ components such that $D(\mathbf{f}, \hat{\mathbf{f}}) \le \epsilon$ with probability $\ge 1 - \delta$, where $D(\cdot, \cdot)$ is some given distance measure, for example $\ell_1$ distance or KL divergence.

An important and extensively studied special case of Gaussian mixtures is mixture of *spherical-Gaussians* [10, 18, 4, 8, 29], where for each component the $d$ coordinates are distributed independently with the same variance, though possibly with different means. Note that different components can have different variances. Due to their simple structure, spherical-Gaussian mixtures are easier to analyze and under a minimum-separation assumption have provably-practical algorithms for clustering and parameter estimation. We consider spherical-Gaussian mixtures as they are important on their own and form a natural first step towards learning general Gaussian mixtures.

## 1.2 Sample complexity

Reducing the number of samples required for learning is of great practical significance. For example, in topic modeling every sample is a whole document, in credit analysis every sample is a person's credit history, and in genetics, every sample is a human DNA. Hence samples can be very scarce and obtaining them can be very costly. By contrast, current CPUs run at several Giga Hertz, hence samples are typically much more scarce of a resource than time.

For one-dimensional distributions, the need for sample-efficient algorithms has been broadly recognized. The sample complexity of many problems is known quite accurately, often to within a constant factor. For example, for discrete distributions over $\{1, \ldots, s\}$, an approach was proposed in [23] and its modifications were used in [28] to estimate the probability multiset using $\Theta(s/\log s)$ samples. Learning one-dimensional $m$-modal distributions over $\{1, \ldots, s\}$ requires $\Theta(m \log(s/m)/\epsilon^3)$ samples [11]. Similarly, one-dimensional mixtures of $k$ structured distributions (log-concave, monotone hazard rate, and unimodal) over $\{1, \ldots, s\}$ can be learned with $\mathcal{O}(k/\epsilon^4)$, $\mathcal{O}(k \log(s/\epsilon)/\epsilon^4)$, and $\mathcal{O}(k \log(s)/\epsilon^4)$ samples, respectively, and these bounds are tight up to a factor of $\epsilon$ [6].

Unlike the 1-dimensional case, in high dimensions, sample complexity bounds are quite weak. For example, to learn a mixture of $k = 2$ spherical Gaussians, existing estimators use $\mathcal{O}(d^{12})$ samples, and this number increases exponentially with $k$ [16]. We close this gap by constructing estimators with near-linear sample complexity.

## 1.3 Previous and new results

Our main contribution is PAC learning $d$-dimensional spherical Gaussian mixtures with near-linear samples. In the process of deriving these results we also prove results for learning one-dimensional Gaussians and for finding which distribution in a class is closest to the one generating samples.

### $d$-dimensional Gaussian mixtures

Several papers considered PAC learning of discrete- and Gaussian-product mixtures. [17] considered mixtures of two $d$-dimensional Bernoulli products where all probabilities are bounded away from 0. They showed that this class is PAC learnable in $\widetilde{\mathcal{O}}(d^2/\epsilon^4)$ time and samples, where the $\widetilde{\mathcal{O}}$ notation hides logarithmic factors. [15] eliminated the probability constraints and generalized the results from binary to arbitrary discrete alphabets and from 2 to $k$ mixture components, showing that these mixtures are PAC learnable in $\widetilde{\mathcal{O}}\big((d/\epsilon)^{2k^2(k+1)}\big)$ time. Although they did not explicitly mention sample complexity, their algorithm uses $\widetilde{\mathcal{O}}\big((d/\epsilon)^{4(k+1)}\big)$ samples. [16] generalized these results to Gaussian products and showed that mixtures of $k$ Gaussians, where the difference between the means is bounded by $B$ times the standard deviation, are PAC learnable in $\widetilde{\mathcal{O}}\big((dB/\epsilon)^{2k^2(k+1)}\big)$ time, and can be shown to use $\widetilde{\mathcal{O}}\big((dB/\epsilon)^{4(k+1)}\big)$ samples. These algorithms consider the KL divergence between the distribution and its estimate, but it can be shown that the $\ell_1$ distance would result in similar complexities. It can also be shown that these algorithms or their simple modifications have similar time and sample complexities for spherical Gaussians as well.

Our main contribution for this problem is to provide an algorithm that PAC learns mixtures of spherical-Gaussians in $\ell_1$ distance with number of samples nearly-linear, and running time polyno-

mial in the dimension $d$. Specifically, in Theorem 11 we show that mixtures of $k$ spherical-Gaussian distributions can be learned using

$$n = \mathcal{O}\left(\frac{dk^9}{\epsilon^4} \log^2 \frac{d}{\delta}\right) = \mathcal{O}_{k,\epsilon}\left(d \log^2 \frac{d}{\delta}\right)$$

samples and in time

$$\mathcal{O}\left(n^2 d \log n + d\left(\frac{k^7}{\epsilon^3} \log^2 \frac{d}{\delta}\right)^{\frac{k^2}{2}}\right) = \widetilde{\mathcal{O}}_{k,\epsilon}(d^3).$$

Recall that for similar problems, previous algorithms used $\widetilde{\mathcal{O}}\left((d/\epsilon)^{4(k+1)}\right)$ samples. Furthermore, recent algorithms typically construct the covariance matrix [29, 16], hence require $\geq nd^2$ time. In that sense, for small $k$, the time complexity we derive is comparable to the best such algorithms one can hope for. Additionally, the exponential dependence on $k$ in the time complexity is $d\left(\frac{k^7}{\epsilon^3} \log^2 \frac{d}{\delta}\right)^{k^2/2}$, significantly lower than the $d^{\mathcal{O}(k^3)}$ dependence in previous results.

Conversely, Theorem 2 shows that any algorithm for PAC learning a mixture of $k$ spherical Gaussians requires $\Omega(dk/\epsilon^2)$ samples, hence our algorithms are nearly sample optimal in the dimension. In addition, their time complexity significantly improves on previously known ones.

**One-dimensional Gaussian mixtures**

To prove the above results we derive two simpler results that are interesting on their own. We construct a simple estimator that learns mixtures of $k$ one-dimensional Gaussians using $\widetilde{\mathcal{O}}(k\epsilon^{-2})$ samples and in time $\widetilde{\mathcal{O}}((k/\epsilon)^{3k+1})$. We note that independently and concurrently with this work [12] showed that mixtures of two one-dimensional Gaussians can be learnt with $\widetilde{\mathcal{O}}(\epsilon^{-2})$ samples and in time $\mathcal{O}(\epsilon^{-5})$. Combining with some of the techniques in this paper, they extend their algorithm to mixtures of $k$ Gaussians, and reduce the exponent to $3k - 1$.

Let $d(\mathbf{f}, \mathcal{F})$ be the smallest $\ell_1$ distance between a distribution $\mathbf{f}$ and any distribution in a collection $\mathcal{F}$. The popular SCHEFFE estimator [13] takes a surprisingly small $\mathcal{O}(\log |\mathcal{F}|)$ independent samples from an unknown distribution $\mathbf{f}$ and time $\mathcal{O}(|\mathcal{F}|^2)$ to find a distribution in $\mathcal{F}$ whose distance from $\mathbf{f}$ is at most a constant factor larger than $d(\mathbf{f}, \mathcal{F})$. In Lemma 1, we reduce the time complexity of the Scheffe algorithm from $\mathcal{O}(|\mathcal{F}|^2)$ to $\widetilde{\mathcal{O}}(|\mathcal{F}|)$, helping us reduce the running time of our algorithms. A detailed analysis of several such estimators are provided in [1] and here we outline a proof for one particular estimator for completeness.

## 1.4  The approach and technical contributions

Given the above, our goal is to construct a small class of distributions such that one of them is $\epsilon$-close to the underlying distribution.

Consider for example mixtures of $k$ components in one dimension with means and variances bounded by $B$. Take the collection of all mixtures derived by quantizing the means and variances of all components to $\epsilon_m$ accuracy, and quantizing the weights to $\epsilon_w$ accuracy. It can be shown that if $\epsilon_m, \epsilon_w \leq \epsilon/k^2$ then one of these candidate mixtures would be $\mathcal{O}(\epsilon)$-close to any mixture, and hence to the underlying one. There are at most $(B/\epsilon_m)^{2k} \cdot (1/\epsilon_w)^k = (B/\epsilon)^{\widetilde{\mathcal{O}}(k)}$ candidates and running SCHEFFE on these mixtures would lead to an estimate. However, this approach requires a bound on the means and variances. We remove this requirement on the bound, by selecting the quantizations based on samples and we describe it in Section 3.

In $d$ dimensions, consider spherical Gaussians with the same variance and means bounded by $B$. Again, take the collection of all distributions derived by quantizing the means of all components in all coordinates to $\epsilon_m$ accuracy, and quantizing the weights to $\epsilon_w$ accuracy. It can be shown that for $d$-dimensional Gaussian to get distance $\epsilon$ from the underlying distribution, it suffices to take $\epsilon_m, \epsilon_w \leq \epsilon^2/\text{poly}(dk)$. There are at most $(B/\epsilon_m)^{dk} \cdot (1/\epsilon_w)^k = 2^{\widetilde{\mathcal{O}}_\epsilon(dk)}$ possible combinations of the $k$ mean vectors and weights. Hence SCHEFFE implies an exponential-time algorithm with sample complexity $\widetilde{\mathcal{O}}(dk)$. To reduce the dependence on $d$, one can approximate the span of the $k$ mean vectors. This reduces the problem from $d$ to $k$ dimensions, allowing us to consider a distribution collection of size $2^{\mathcal{O}(k^2)}$, with SCHEFFE sample complexity of just $\mathcal{O}(k^2)$. [15, 16] constructs the sample correlation matrix and uses $k$ of its columns to approximate the span of mean vectors. This

approach requires the $k$ columns of the sample correlation matrix to be very close to the actual correlation matrix, requiring a lot more samples.

We derive a spectral algorithm that approximates the span of the $k$ mean vectors using the top $k$ eigenvectors of the sample covariance matrix. Since we use the entire covariance matrix instead of just $k$ columns, a weaker concentration suffices and the sample complexity can be reduced.

Using recent tools from non-asymptotic random matrix theory [30], we show that the span of the means can be approximated with $\widetilde{\mathcal{O}}(d)$ samples. This result allows us to address most "reasonable" distributions, but still there are some "corner cases" that need to be analyzed separately. To address them, we modify some known clustering algorithms such as single-linkage, and spectral projections. While the basic algorithms were known before, our contribution here, which takes a fair bit of effort and space, is to show that judicious modifications of the algorithms and rigorous statistical analysis yield polynomial time algorithms with near-linear sample complexity. We provide a simple and practical spectral algorithm that estimates all such mixtures in $\mathcal{O}_{k,\epsilon}(d \log^2 d)$ samples.

The paper is organized as follows. In Section 2, we introduce notations, describe results on the Scheffe estimator, and state a lower bound. In Sections 3 and 4, we present the algorithms for one-dimensional and $d$-dimensional Gaussian mixtures respectively. Due to space constraints, most of the technical details and proofs are given in the appendix.

## 2 Preliminaries

### 2.1 Notation

For arbitrary product distributions $\mathbf{p}_1, \ldots, \mathbf{p}_k$ over a $d$ dimensional space let $p_{j,i}$ be the distribution of $\mathbf{p}_j$ over coordinate $i$, and let $\mu_{j,i}$ and $\sigma_{j,i}$ be the mean and variance of $p_{j,i}$ respectively. Let $\mathbf{f} = (w_1, \ldots, w_k, \mathbf{p}_1, \ldots, \mathbf{p}_k)$ be the mixture of these distributions with mixing weights $w_1, \ldots, w_k$. We denote estimates of a quantity $\mathbf{x}$ by $\hat{\mathbf{x}}$. It can be empirical mean or a more complex estimate. $\|\cdot\|$ denotes the spectral norm of a matrix and $\|\cdot\|_2$ is the $\ell_2$ norm of a vector. We use $D(\cdot, \cdot)$ to denote the $\ell_1$ distance between two distributions.

### 2.2 Selection from a pool of distributions

Many algorithms for learning mixtures over the domain $\mathcal{X}$ first obtain a *small* collection $\mathcal{F}$ of mixtures and then perform Maximum Likelihood test using the samples to output a distribution [15, 17]. Our algorithm also obtains a set of distributions containing at least one that is close to the underlying in $\ell_1$ distance. The estimation problem now reduces to the following. Given a class $\mathcal{F}$ of distributions and samples from an unknown distribution $\mathbf{f}$, find a distribution in $\mathcal{F}$ that is *close* to $\mathbf{f}$. Let $D(\mathbf{f}, \mathcal{F}) \overset{\text{def}}{=} \min_{\mathbf{f}_i \in \mathcal{F}} D(\mathbf{f}, \mathbf{f}_i)$.

The well-known Scheffe's method [13] uses $\mathcal{O}(\epsilon^{-2} \log |\mathcal{F}|)$ samples from the underlying distribution $\mathbf{f}$, and in time $\mathcal{O}(\epsilon^{-2} |\mathcal{F}|^2 T \log |\mathcal{F}|)$ outputs a distribution in $\mathcal{F}$ with $\ell_1$ distance of at most $9.1 \cdot \max(D(\mathbf{f}, \mathcal{F}), \epsilon)$ from $\mathbf{f}$, where $T$ is the time required to compute the probability of an $x \in \mathcal{X}$ by a distribution in $\mathcal{F}$. A naive application of this algorithm requires time quadratic in the number of distributions in $\mathcal{F}$. We propose a variant of this, that works in near linear time. More precisely,

**Lemma 1** (Appendix B). *Let $\epsilon > 0$. For some constant $c$, given $\frac{c}{\epsilon^2} \log\left(\frac{|\mathcal{F}|}{\delta}\right)$ independent samples from a distribution $\mathbf{f}$, with probability $\geq 1 - \delta$, the output $\hat{\mathbf{f}}$ of* MODIFIED SCHEFFE *satisfies $D(\hat{\mathbf{f}}, \mathbf{f}) \leq 1000 \cdot \max(D(\mathbf{f}, \mathcal{F}), \epsilon)$. Furthermore, the algorithm runs in time $\mathcal{O}\left(\frac{|\mathcal{F}| T \log(|\mathcal{F}|/\delta)}{\epsilon^2}\right)$.*

Several such estimators have been proposed in the past [11, 12]. A detailed analysis of the estimator presented here was studied in [1]. We outline a proof in Appendix B for completeness. Note that the constant 1000 in the above lemma has not been optimized. For our problem of estimating $k$ component mixtures in $d$-dimensions, $T = \mathcal{O}(dk)$ and $|\mathcal{F}| = \widetilde{\mathcal{O}}_{k,\epsilon}(d^2)$.

### 2.3 Lower bound

Using Fano's inequality, we show an information theoretic lower bound of $\Omega(dk/\epsilon^2)$ samples to learn $k$-component $d$-dimensional spherical Gaussian mixtures for any algorithm. More precisely,

**Theorem 2** (Appendix C). *Any algorithm that learns all $k$-component $d$-dimensional spherical Gaussian mixtures to $\ell_1$ distance $\epsilon$ with probability $\geq 1/2$ requires $\Omega(dk/\epsilon^2)$ samples.*

## 3 Mixtures in one dimension

Over the past decade estimation of one dimensional distributions has gained significant attention [24, 28, 11, 6, 12, 7]. We provide a simple estimator for learning one dimensional Gaussian mixtures using the MODIFIED SCHEFFE estimator. Formally, given samples from $f$, a mixture of Gaussian distributions $p_i \stackrel{\text{def}}{=} N(\mu_i, \sigma_i^2)$ with weights $w_1, w_2, \ldots w_k$, our goal is to find a mixture $\hat{f} = (\hat{w}_1, \hat{w}_2, \ldots \hat{w}_k, \hat{p}_1, \hat{p}_2, \ldots \hat{p}_k)$ such that $D(f, \hat{f}) \leq \epsilon$. We make no assumption on the weights, means or the variances of the components. While we do not use the one dimensional algorithm in the $d$-dimensional setting, it provides insight to the usage of the MODIFIED SCHEFFE estimator and may be of independent interest. As stated in Section 1.4, our quantizations are based on samples and is an immediate consequence of the following observation for samples from a Gaussian distribution.

**Lemma 3** (Appendix D.1). *Given $n$ independent samples $x_1, \ldots, x_n$ from $N(\mu, \sigma^2)$, with probability $\geq 1 - \delta$ there are two samples $x_j, x_k$ such that $|x_j - \mu| \leq \sigma \frac{7 \log 2/\delta}{2n}$ and $|x_j - x_k - \sigma| \leq 2\sigma \frac{7 \log 2/\delta}{2n}$.*

The above lemma states that given samples from a Gaussian distribution, there would be a sample close to the mean and there would be two samples that are about a standard deviation apart. Hence, if we consider the set of all Gaussians $N(x_j, (x_j - x_k)^2) : 1 \leq j, k \leq n$, then that set would contain a Gaussian close to the underlying one. The same holds for mixtures and for a Gaussian mixture and we can create the set of candidate mixtures as follows.

**Lemma 4** (Appendix D.2). *Given $n \geq \frac{120k \log(4k/\delta)}{\epsilon}$ samples from a mixture $f$ of $k$ Gaussians. Let $S = \{N(x_j, (x_j - x_k)^2) : 1 \leq j, k \leq n\}$ and $W = \{0, \frac{\epsilon}{2k}, \frac{2\epsilon}{2k} \ldots, 1\}$ be a set of weights. Let*

$$\mathcal{F} \stackrel{\text{def}}{=} \{(\hat{w}_1, \hat{w}_2, \ldots, \hat{w}_k, \hat{p}_1, \hat{p}_2, \ldots \hat{p}_k) : \hat{p}_i \in S, \forall 1 \leq i \leq k-1, \hat{w}_i \in W, \hat{w}_k = 1 - (\hat{w}_1 + \ldots \hat{w}_{k-1}) \geq 0\}$$

*be a set of $n^{2k}(2k/\epsilon)^{k-1} \leq n^{3k-1}$ candidate distributions. There exists $\hat{f} \in \mathcal{F}$ such that $D(f, \hat{f}) \leq \epsilon$.*

Running the MODIFIED SCHEFFE algorithm on the above set of candidates $\mathcal{F}$ yields a mixture that is close to the underlying one. By Lemma 1 and the above lemma we obtain

**Corollary 5** (Appendix D.3). *Let $n \geq c \cdot \frac{k}{\epsilon^2} \log \frac{k}{\epsilon\delta}$ for some constant $c$. There is an algorithm that runs in time $\mathcal{O}\left(\left(\frac{k \log(k/\epsilon\delta)}{\epsilon}\right)^{3k-1} \frac{k^2 \log(k/\epsilon\delta)}{\epsilon^2}\right)$, and returns a mixture $\hat{f}$ such that $D(f, \hat{f}) \leq 1000\epsilon$ with probability $\geq 1 - 2\delta$.*

[12] considered the one dimensional Gaussian mixture problem for two component mixtures. While the process of identifying the candidate means is same for both the papers, the process of identifying the variances and proof techniques are different.

## 4 Mixtures in $d$ dimensions

Algorithm LEARN $k$-SPHERE learns mixtures of $k$ spherical Gaussians using near-linear samples. For clarity and simplicity of proofs, we first prove the result when all components have the same variance $\sigma^2$, *i.e.*, $\mathbf{p}_i = N(\boldsymbol{\mu}_i, \sigma^2 \mathbb{I}_d)$ for $1 \leq i \leq k$. A modification of this algorithm works for components with different variances. The core ideas are same and we discuss the changes in Section 4.3. The algorithm starts out by estimating $\sigma^2$ and we discuss this step later. We estimate the means in three steps, a coarse single-linkage clustering, recursive spectral clustering and search over span of means. We now discuss the necessity of these steps.

### 4.1 Estimating the span of means

A simple modification of the one dimensional algorithm can be used to learn mixtures in $d$ dimensions, however, the number of candidate mixtures would be exponential in $d$, the number of dimensions. As stated in Section 1.4, given the span of the mean vectors $\boldsymbol{\mu}_i$, we can grid the $k$ dimensional span to the required accuracy $\epsilon_g$ and use MODIFIED SCHEFFE, to obtain a polynomial

time algorithm. One of the natural and well-used methods to estimate the span of mean vectors is using the correlation matrix [29]. Consider the correlation-type matrix,

$$S = \frac{1}{n} \sum_{i=1}^{n} \mathbf{X}(i)\mathbf{X}(i)^t - \sigma^2 \mathbb{I}_d.$$

For a sample $\mathbf{X}$ from a particular component $j$, $\mathbb{E}[\mathbf{X}\mathbf{X}^t] = \sigma^2 \mathbb{I}_d + \boldsymbol{\mu}_j {\boldsymbol{\mu}_j}^t$, and the expected fraction of samples from $\mathbf{p}_j$ is $w_j$. Hence

$$\mathbb{E}[S] = \sum_{j=1}^{k} w_j \boldsymbol{\mu}_j {\boldsymbol{\mu}_j}^t.$$

Therefore, as $n \to \infty$, $S$ converges to $\sum_{j=1}^{k} w_j \boldsymbol{\mu}_j {\boldsymbol{\mu}_j}^t$, and its top $k$ eigenvectors span the means.

While the above intuition is well understood, the number of samples necessary for convergence is not well studied. We wish $\widetilde{\mathcal{O}}(d)$ samples to be sufficient for the convergence irrespective of the values of the means. However this is not true when the means are far apart. In the following example we demonstrate that the convergence of averages can depend on their separation.

**Example 6.** *Consider the special case, $d = 1$, $k = 2$, $\sigma^2 = 1$, $w_1 = w_2 = 1/2$, and mean differences $|\mu_1 - \mu_2| = L \gg 1$. Given this prior information, one can estimate the average of the mixture, that yields $(\mu_1 + \mu_2)/2$. Solving equations obtained by $\mu_1 + \mu_2$ and $\mu_1 - \mu_2 = L$ yields $\mu_1$ and $\mu_2$. The variance of the mixture is $1 + L^2/4 > L^2/4$. With additional Chernoff type bounds, one can show that given $n$ samples the error in estimating the average is*

$$|\mu_1 + \mu_2 - \hat{\mu}_1 - \hat{\mu}_2| \approx \Theta\left(L/\sqrt{n}\right).$$

*Hence, estimating the means to high precision requires $n \geq L^2$, i.e., the higher separation, the more samples are necessary if we use the sample mean.*

A similar phenomenon happens in the convergence of the correlation matrices, where the variances of quantities of interest increase with separation. In other words, for the span to be accurate the number of samples necessary increases with the separation. To overcome this, a natural idea is to cluster the Gaussians such that the component means in the same cluster are close and then estimate the span of means, and apply SCHEFFE on the span within each cluster.

For clustering, we use another spectral algorithm. Even though spectral clustering algorithms are studied in [29, 2], they assume that the weights are strictly bounded away from 0, which does not hold here. We use a simple recursive clustering algorithm that takes a cluster $C$ with average $\overline{\boldsymbol{\mu}}(C)$. If there is a component in the cluster such that $\sqrt{w_i} \|\boldsymbol{\mu}_i - \overline{\boldsymbol{\mu}}(C)\|_2$ is $\Omega(\log(n/\delta)\sigma)$, then the algorithm divides the cluster into two nonempty clusters without any mis-clustering. For technical reasons similar to the above example, we first use a coarse clustering algorithm that ensures that the mean separation of any two components within each cluster is $\widetilde{\mathcal{O}}(d^{1/4}\sigma)$.

Our algorithm thus comprises of $(i)$ variance estimation $(ii)$ a coarse clustering ensuring that means are within $\widetilde{\mathcal{O}}(d^{1/4}\sigma)$ of each other in each cluster $(iii)$ a recursive spectral clustering that reduces the mean separation to $\mathcal{O}(\sqrt{k^3 \log(n/\delta)}\sigma)$ $(iv)$ estimating the span of mean within each cluster, and $(v)$ quantizing the means and running MODIFIED SCHFEE on the resulting candidate mixtures.

## 4.2 Sketch of correctness

We now describe the steps stating the performance of each step of Algorithm LEARN $k$-SPHERE. To simplify the bounds and expressions, we assume that $d > 1000$ and $\delta \geq \min(2n^2 e^{-d/10}, 1/3)$. For smaller values of $\delta$, we run the algorithm with error $1/3$ and repeat it $\mathcal{O}(\log \frac{1}{\delta})$ times to choose a set of candidate mixtures $\mathcal{F}_\delta$. By the Chernoff-bound with error $\leq \delta$, $\mathcal{F}_\delta$ contains a mixture $\epsilon$-close to $\mathbf{f}$. Finally, we run MODIFIED SCHEFFE on $\mathcal{F}_\delta$ to obtain a mixture that is close to $\mathbf{f}$. By the union bound and Lemma 1, the error of the new algorithm is $\leq 2\delta$.

**Variance estimation:** Let $\hat{\sigma}$ be the variance estimate from step 1. If $\mathbf{X}(1)$ and $\mathbf{X}(2)$ are two samples from the components $i$ and $j$ respectively, then $\mathbf{X}(1) - \mathbf{X}(2)$ is distributed $N(\boldsymbol{\mu}_i - \boldsymbol{\mu}_j, 2\sigma^2 \mathbb{I}_d)$. Hence for large $d$, $\|\mathbf{X}(1) - \mathbf{X}(2)\|_2^2$ concentrates around $2d\sigma^2 + \|\boldsymbol{\mu}_i - \boldsymbol{\mu}_j\|_2^2$. By the pigeon-hole principle, given $k + 1$ samples, two of them are from the same component. Therefore, the minimum pairwise

distance between $k + 1$ samples is close to $2d\sigma^2$. This is made precise in the next lemma which states that $\hat{\sigma}^2$ is a good estimate of the variance.

**Lemma 7** (Appendix E.1). *Given $n$ samples from the $k$-component mixture, with probability $1 - 2\delta$, $|\hat{\sigma}^2 - \sigma^2| \leq 2.5\sigma^2\sqrt{\log(n^2/\delta)/d}$.*

**Coarse single-linkage clustering:** The second step is a single-linkage routine that clusters mixture components with *far* means. Single-linkage is a simple clustering scheme that starts out with each data point as a cluster, and at each step merges the two nearest clusters to form a larger cluster. The algorithm stops when the distance between clusters is larger than a pre-specified threshold.

Suppose the samples are generated by a one-dimensional mixture of $k$ components that are *far*, then *with high probability*, when the algorithm generates $k$ clusters all the samples within a cluster are generated by a single component. More precisely, if $\forall i, j \in [k]$, $|\mu_i - \mu_j| = \Omega(\sigma \log n)$, then all the $n$ samples concentrate around their respective means and the separation between any two samples from different components would be larger than the largest separation between any two samples from the same component. Hence for a suitable value of threshold, single-linkage correctly identifies the clusters. For $d$-dimensional Gaussian mixtures a similar property holds, with minimum separation $\Omega((d \log \frac{n}{\delta})^{1/4}\sigma)$. More precisely,

**Lemma 8** (Appendix E.2). *After Step 2 of* LEARN $k$-SPHERE*, with probability $\geq 1 - 2\delta$, all samples from each component will be in the same cluster and the maximum distance between two components within each cluster is $\leq 10k\sigma\left(d \log \frac{n^2}{\delta}\right)^{1/4}$.*

---

**Algorithm** LEARN $k$-SPHERE
**Input:** $n$ samples $\mathbf{x}(1), \mathbf{x}(2), \ldots, \mathbf{x}(n)$ from $\mathbf{f}$ and $\epsilon$.

1. **Sample variance:** $\hat{\sigma}^2 = \min_{a \neq b: a, b \in [k+1]} \|\mathbf{x}(a) - \mathbf{x}(b)\|_2^2 / 2d$.

2. **Coarse single-linkage clustering:** Start with each sample as a cluster,

   - While $\exists$ two clusters with squared-distance $\leq 2d\hat{\sigma}^2 + 23\hat{\sigma}^2\sqrt{d \log(n^2/\delta)}$, merge them.

3. **Recursive spectral-clustering:** While there is a cluster $C$ with $|C| \geq n\epsilon/5k$ and spectral norm of its sample covariance matrix $\geq 12k^2\hat{\sigma}^2 \log n^3/\delta$,

   - Use $n\epsilon/8k^2$ of the samples to find the largest eigenvector and discard these samples.
   - Project the remaining samples on the largest eigenvector.
   - Perform single-linkage in the projected space (as before) till the distance between clusters is $> 3\hat{\sigma}\sqrt{\log(n^2k/\delta)}$ creating new clusters.

4. **Exhaustive search**: Let $\epsilon_g = \epsilon/(16k^{3/2})$, $L = 200\sqrt{k^4\epsilon^{-1} \log \frac{n^2}{\delta}}$, $L' = \frac{32k\sqrt{\log n^2/\delta}}{\epsilon}$, and $G = \{-L, \ldots, -\epsilon_g, 0, \epsilon_g, 2\epsilon_g, \ldots L\}$. Let $W = \{0, \epsilon/(4k), 2\epsilon/(4k), \ldots 1\}$ and $\Sigma \stackrel{\text{def}}{=} \{\sigma^2 : \sigma^2 = \hat{\sigma}^2(1 + i\epsilon/d\sqrt{128dk^2}), \forall -L' < i \leq L'\}$.

   - For each cluster $C$ find its top $k - 1$ eigenvectors $\mathbf{u}_1, \ldots \mathbf{u}_{k-1}$. Let $\text{Span}(C) = \{\hat{\boldsymbol{\mu}}(C) + \sum_{i=1}^{k-1} g_i\hat{\sigma}\mathbf{u}_i : g_i \in G\}$.
   - Let $\text{Span} = \cup_{C:|C| \geq \frac{n\epsilon}{5k}} \text{Span}(C)$.
   - For all $w_i' \in W$, $\sigma'^2 \in \Sigma$, $\hat{\boldsymbol{\mu}}_i \in \text{Span}$,
     add $\{(w_1', \ldots, w_{k-1}', 1 - \sum_{i=1}^{k-1} w_i', N(\hat{\boldsymbol{\mu}}_1, \sigma'^2), \ldots, N(\hat{\boldsymbol{\mu}}_k, \sigma'^2))\}$ to $\mathcal{F}$.

5. Run MODIFIED SCHEFFE on $\mathcal{F}$ and output the resulting distribution.

---

**Recursive spectral-clustering:** The clusters formed at the beginning of this step consist of components with mean separation $\mathcal{O}(\sigma d^{1/4} \log \frac{n}{\delta})$. We now recursively zoom into the clusters formed and show that it is possible to cluster the components with much smaller mean separation. Note that since the matrix is symmetric, the largest magnitude of the eigenvalue is the same as the spectral norm. We first find the largest eigenvector of

$$S(C) \stackrel{\text{def}}{=} \frac{1}{|C|}\left(\sum_{\mathbf{x} \in C} (\mathbf{x} - \hat{\boldsymbol{\mu}}(C))(\mathbf{x} - \hat{\boldsymbol{\mu}}(C))^t\right) - \hat{\sigma}^2 \mathbb{I}_d,$$

which is the sample covariance matrix with its diagonal term reduced by $\hat{\sigma}^2$. We then project our samples to this vector and if there are two components with means far apart, then using single-linkage we divide the cluster into two. The following lemma shows that this step performs accurate clustering of components with well separated means.

**Lemma 9** (Appendix E.3). *Let* $n \geq c \cdot \frac{dk^4}{\epsilon} \log \frac{n^3}{\delta}$. *After recursive clustering, with probability* $\geq 1 - 4\delta$, *the samples are divided into clusters such that for each component* $i$ *within a cluster* $C$, $\sqrt{w_i} \|\boldsymbol{\mu}_i - \overline{\boldsymbol{\mu}}(C)\|_2 \leq 25\sigma\sqrt{k^3 \log(n^3/\delta)}$ . *Furthermore, all the samples from one component remain in a single cluster.*

**Exhaustive search and Scheffe:** After step 3, all clusters have a small weighted radius $\sqrt{w_i} \|\boldsymbol{\mu}_i - \overline{\boldsymbol{\mu}}(C)\|_2 \leq 25\sigma\sqrt{k^3 \log \frac{n^3}{\delta}}$. It can be shown that the eigenvectors give an accurate estimate of the span of $\boldsymbol{\mu}_i - \overline{\boldsymbol{\mu}}(C)$ within each cluster. More precisely,

**Lemma 10** (Appendix E.4). *Let* $n \geq c \cdot \frac{dk^9}{\epsilon^4} \log^2 \frac{d}{\delta}$ *for some constant c. After step 3, with probability* $\geq 1 - 7\delta$, *if* $|C| \geq n\epsilon/5k$, *then the projection of* $[\boldsymbol{\mu}_i - \overline{\boldsymbol{\mu}}(C)]/\|\boldsymbol{\mu}_i - \overline{\boldsymbol{\mu}}(C)\|_2$ *on the space orthogonal to the span of top* $k - 1$ *eigenvectors has magnitude* $\leq \frac{\epsilon\sigma}{8\sqrt{2}k\sqrt{w_i}\|\boldsymbol{\mu}_i - \overline{\boldsymbol{\mu}}(C)\|_2}$.

We now have accurate estimates of the spans of the cluster means and each cluster has components with close means. It is now possible to grid the set of possibilities in each cluster to obtain a set of distributions such that one of them is close to the underlying. There is a trade-off between a dense grid to obtain a good estimation and the computation time required. The final step takes the sparsest grid possible to ensure an error $\leq \epsilon$. This is quantized below.

**Theorem 11** (Appendix E.5). *Let* $n \geq c \cdot \frac{dk^9}{\epsilon^4} \log^2 \frac{d}{\delta}$ *for some constant c. Then Algorithm* LEARN $k$-SPHERE, *with probability* $\geq 1 - 9\delta$, *outputs a distribution* $\hat{\mathbf{f}}$ *such that* $D(\hat{\mathbf{f}}, \mathbf{f}) \leq 1000\epsilon$. *Furthermore, the algorithm runs in time* $\mathcal{O}\left(n^2 d \log n + d\left(\frac{k^7}{\epsilon^3} \log^2 \frac{d}{\delta}\right)^{\frac{k^2}{2}}\right)$.

Note that the run time is calculated based on an efficient implementation of single-linkage clustering and the exponential term is not optimized.

## 4.3 Mixtures with unequal variances

We generalize the results to mixtures with components having different variances. Let $\mathbf{p}_i = N(\boldsymbol{\mu}_i, \sigma_i^2 \mathbb{I}_d)$ be the $i$th component. The key differences between LEARN $k$-SPHERE and the algorithm for learning mixtures with unequal variances are:

1. In LEARN $k$-SPHERE, we first estimated the component variance $\sigma$ and divided the samples into clusters such that within each cluster the means are separated by $\widetilde{\mathcal{O}}(d^{1/4}\sigma)$. We modify this step such that the samples are clustered such that within each cluster the components not only have mean separation $\mathcal{O}(d^{1/4}\sigma)$, but variances are also a factor at most $1 + \widetilde{\mathcal{O}}(1/\sqrt{d})$ apart.

2. Once the variances in each cluster are within a multiplicative factor of $1 + \widetilde{\mathcal{O}}(1/\sqrt{d})$ of each other, it can be shown that the performance of the recursive spectral clustering step does not change more than constants.

3. After obtaining clusters with *similar* means and variances, the exhaustive search algorithm follows, though instead of having a single $\sigma'$ for all clusters, we can have a different $\sigma'$ for each cluster, which is estimated using the average pair wise distance between samples in the cluster.

The changes in the recursive clustering step and the exhaustive search step are easy to see and we omit them. The coarse clustering step requires additional tools and we describe them in Appendix F.

## 5 Acknowledgements

We thank Sanjoy Dasgupta, Todd Kemp, and Krishnamurthy Vishwanathan for helpful discussions.

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
