[Supplementary Material]

# A  Useful tools

## A.1  Bounds on $\ell_1$ distance

For two $d$ dimensional product distributions $\mathbf{p}_1$ and $\mathbf{p}_2$, if we bound the $\ell_1$ distance on each coordinate by $\epsilon$, then by triangle inequality $D(\mathbf{p}_1, \mathbf{p}_2) \leq d\epsilon$. However this bound is often weak. One way to obtain a stronger bound is to relate $\ell_1$ distance to Bhattacharyya parameter, which is defined as follows: Bhattacharyya parameter $B(p_1, p_2)$ between two distributions $p_1$ and $p_2$ is

$$B(p_1, p_2) = \int_{x \in \mathcal{X}} \sqrt{p_1(x) p_2(x)} dx.$$

The $\ell_1$ distance between $p_1$ and $p_2$ can be bounded in terms of $B(p_1, p_2)$ as follows.

**Lemma 12.** *For distributions $p_1$ and $p_2$,*

$$D(p_1, p_2)^2 \leq 8(1 - B(p_1, p_2)).$$

*Proof.* Since $\int_{x \in \mathcal{X}} p_1(x) dx = \int_{x \in \mathcal{X}} p_2(x) dx = 1$,

$$\int_{x \in \mathcal{X}} \left( \sqrt{p_1(x)} - \sqrt{p_2(x)} \right)^2 dx = 2(1 - B(p_1, p_2)).$$

Moreover since $(a + b)^2 \leq 2a^2 + 2b^2$,

$$\int_{x \in \mathcal{X}} \left( \sqrt{p_1(x)} + \sqrt{p_2(x)} \right)^2 dx \leq 4$$

Using these bounds with the following Cauchy-Schwarz inequality proves the lemma.

$$\int_{x \in \mathcal{X}} \left( \sqrt{p_1(x)} + \sqrt{p_2(x)} \right)^2 dx \cdot \int_{x \in \mathcal{X}} \left( \sqrt{p_1(x)} - \sqrt{p_2(x)} \right)^2 dx \geq \left( \int_{x \in \mathcal{X}} |p_1(x) - p_2(x)| dx \right)^2 = D(p_1, p_2)^2.$$

$\square$

By the definition of Bhattacharyya distance, it is multiplicative for product distributions, namely for two product distributions $\mathbf{p}_1$ and $\mathbf{p}_2$, $B(\mathbf{p}_1, \mathbf{p}_2) = \prod_{i=1}^{d} B(p_{1,i}, p_{2,i})$. We use this with the previous lemma to bound the $\ell_1$ distance of Gaussian mixtures.

We first bound Bhattacharyya parameter for two one-dimensional Gaussian distributions.

**Lemma 13.** *The Bhattacharyya parameter for two one dimensional Gaussian distributions $p_1 = N(\mu_1, \sigma_1^2)$ and $p_2 = N(\mu_2, \sigma_2^2)$ is*

$$B(p_1, p_2) \geq 1 - \frac{(\mu_1 - \mu_2)^2}{4(\sigma_1^2 + \sigma_2^2)} - \frac{(\sigma_1^2 - \sigma_2^2)^2}{(\sigma_1^2 + \sigma_2^2)^2}.$$

*Proof.* For Gaussian distributions a straight-forward computation shows that $B(p_1, p_2) = y e^{-x}$, where $x = \frac{(\mu_1 - \mu_2)^2)}{4(\sigma_1^2 + \sigma_2^2)}$ and $y = \sqrt{\frac{2\sigma_1 \sigma_2}{\sigma_1^2 + \sigma_2^2}}$. Observe that

$$y = \sqrt{\frac{2\sigma_1 \sigma_2}{\sigma_1^2 + \sigma_2^2}} = \sqrt{1 - \frac{(\sigma_1 - \sigma_2)^2}{\sigma_1^2 + \sigma_2^2}} \geq 1 - \frac{(\sigma_1 - \sigma_2)^2}{\sigma_1^2 + \sigma_2^2} \geq 1 - \frac{(\sigma_1^2 - \sigma_2^2)^2}{(\sigma_1^2 + \sigma_2^2)^2}.$$

Hence,

$$B(p_1, p_2) = y e^{-x} \geq y(1 - x) \geq (1 - x)\left(1 - \frac{(\sigma_1^2 - \sigma_2^2)^2}{(\sigma_1^2 + \sigma_2^2)^2}\right) \geq 1 - x - \frac{(\sigma_1^2 - \sigma_2^2)^2}{(\sigma_1^2 + \sigma_2^2)^2}.$$

Substituting the value of $x$ results in the lemma. $\square$

Therefore,

$$\begin{aligned}
B(\mathbf{p}_1, \mathbf{p}_2) &= \prod_{i=1}^{d} B(p_{1,i}, p_{2,i}) \\
&\geq \prod_{i=1}^{d} \left[ 1 - \frac{(\mu_{1,i} - \mu_{2,i})^2}{4(\sigma_{1,i}^2 + \sigma_{2,i}^2)} - \frac{(\sigma_{1,i}^2 - \sigma_{2,i}^2)^2}{(\sigma_{1,i}^2 + \sigma_{2,i}^2)^2} \right] \\
&\geq 1 - \sum_{i=1}^{d} \left[ \frac{(\mu_{1,i} - \mu_{2,i})^2}{4(\sigma_{1,i}^2 + \sigma_{2,i}^2)} + \frac{(\sigma_{1,i}^2 - \sigma_{2,i}^2)^2}{(\sigma_{1,i}^2 + \sigma_{2,i}^2)^2} \right],
\end{aligned}$$

where the last step uses $\prod(1 - x_i) \geq 1 - \sum_i x_i$ for $x_i \in (0, 1)$.

Using this with Lemma 12,

**Lemma 14.** *For any two Gaussian product distributions* $\mathbf{p}_1$ *and* $\mathbf{p}_2$,

$$D(\mathbf{p}_1, \mathbf{p}_2)^2 \leq \sum_{i=1}^{d} 2\frac{(\mu_{1,i} - \mu_{2,i})^2}{\sigma_{1,i}^2 + \sigma_{2,i}^2} + 8\frac{(\sigma_{1,i}^2 - \sigma_{2,i}^2)^2}{(\sigma_{1,i}^2 + \sigma_{2,i}^2)^2}. \qquad \square$$

## A.2 Concentration inequalities

We use the following concentration inequalities for Gaussian, Chi-Square, and sum of Bernoulli random variables in the rest of the paper.

**Lemma 15.** *For a Gaussian random variable* $X$ *with mean* $\mu$ *and variance* $\sigma^2$,

$$\Pr(|X - \mu| \geq t\sigma) \leq e^{-t^2/2}.$$

**Lemma 16** (Chi-square bounds). *If* $Y_1, Y_2, \ldots Y_n$ *be* $n$ *i.i.d. Gaussian variables with mean* $0$ *and variance* $\sigma^2$, *then*

$$\Pr\left(\sum_{i=1}^{n} Y_i^2 - n\sigma^2 \geq 2(\sqrt{nt} + t)\sigma^2\right) \leq e^{-t}, \text{ and } \Pr\left(\sum_{i=1}^{n} Y_i^2 - n\sigma^2 \leq -2\sqrt{nt}\sigma^2\right) \leq e^{-t}.$$

*Furthermore for a fixed vector* $\mathbf{a}$,

$$\Pr\left(\left|\sum_{i=1}^{n} \mathbf{a}_i(Y_i^2 - 1)\right| \leq 2(\|\mathbf{a}\|_2 \sqrt{t} + \|\mathbf{a}\|_\infty t)\sigma^2\right) \leq 2e^{-t}.$$

A simple combination of the above two results proves the following.

**Lemma 17.** *If* $\mathbf{X}$ *is distributed according to* $N(\boldsymbol{\mu}, \sigma^2 \mathbb{I}_d)$ *then,*

$$\Pr\left(-2\sqrt{dt}\sigma^2 - 2\|\boldsymbol{\mu}\|_2 t\sigma \geq \|\mathbf{X}\|_2^2 - \|\boldsymbol{\mu}\|_2^2 - d\sigma^2 \geq 2(\sqrt{dt} + t)\sigma^2 + 2\|\boldsymbol{\mu}\|_2 t\sigma\right) \leq 2e^{-t} + e^{-t^2/2}.$$

**Lemma 18** (Chernoff bound). *If* $X_1, X_2 \ldots X_n$ *are distributed according to Bernoulli* $p$, *then with probability* $1 - \delta$,

$$\left|\frac{\sum_{i=1}^{n} X_i}{n} - p\right| \leq \sqrt{\frac{2p(1-p)}{n} \log \frac{2}{\delta}} + \frac{2}{3} \frac{\log \frac{2}{\delta}}{n}.$$

We now state a non-asymptotic concentration inequality for random matrices that helps us bound errors in spectral algorithms.

**Lemma 19** ([30] Remark 5.51). *Let* $\mathbf{y}(1), \mathbf{y}(2), \ldots, \mathbf{y}(n)$ *be generated according to* $N(0, \Sigma)$. *For every* $\epsilon \in (0, 1)$ *and* $t \geq 1$, *if* $n \geq c'd\left(\frac{t}{\epsilon}\right)^2$ *for some constant* $c'$, *then with probability* $\geq 1 - 2e^{-t^2 n}$,

$$\left\|\sum_{i=1}^{n} \frac{1}{n}\mathbf{y}(i)\mathbf{y}^t(i) - \Sigma\right\| \leq \epsilon \|\Sigma\|.$$

## A.3 Matrix eigenvalues

We now state few simple lemmas on the eigenvalues of perturbed matrices.

**Lemma 20.** *Let* $\lambda_1^A \geq \lambda^A \geq \ldots \lambda_d^A \geq 0$ *and* $\lambda_1^B \geq \lambda^B \geq \ldots \lambda_d^B \geq 0$ *be the eigenvalues of two symmetric matrices* $A$ *and* $B$ *respectively. If* $\|A - B\| \leq \epsilon$, *then* $\forall i, |\lambda_i^A - \lambda_i^B| \leq \epsilon$.

*Proof.* Let $\mathbf{u}_1, \mathbf{u}_2, \ldots \mathbf{u}_d$ be a set of eigenvectors of $A$ that corresponds to $\lambda_1^A, \lambda_2^A, \ldots \lambda_d^A$. Similarly let $\mathbf{v}_1, \mathbf{v}_2, \ldots \mathbf{v}_d$ be eigenvectors of $B$ Consider the first eigenvalue of $B$,

$$\lambda_1^B = \|B\| = \|A + (B - A)\| \geq \|A\| - \|B - A\| \geq \lambda_1^A - \epsilon.$$

Now consider an $i > 1$. If $\lambda_i^B < \lambda_i^A - \epsilon$, then by definition of eigenvalues

$$\max_{\mathbf{v}: \forall j \leq i-1, \mathbf{v} \cdot \mathbf{v}_j = 0} \|B\mathbf{v}\|_2 < \lambda_i^A - \epsilon.$$

Now consider a unit vector $\sum_{j=1}^{i} \alpha_j \mathbf{u}_j$ in the span of $\mathbf{u}_1, \ldots \mathbf{u}_i$, that is orthogonal to $\mathbf{v}_1, \ldots \mathbf{v}_{i-1}$. For this vector,

$$\left\| B \sum_{j=1}^{i} \alpha_j \mathbf{u}_j \right\|_2 \geq \left\| A \sum_{j=1}^{i} \alpha_j \mathbf{u}_j \right\|_2 - \left\| (A-B) \sum_{j=1}^{i} \alpha_j \mathbf{u}_j \right\|_2 \geq \sqrt{\sum_{j=1}^{i} \alpha_j^2 (\lambda_j^A)^2} - \epsilon \geq \lambda_i^A - \epsilon,$$

a contradiction. Hence, $\forall i \leq d$, $\lambda_i^B \geq \lambda_i^A - \epsilon$. The proof in the other direction is similar and omitted. $\qquad \square$

**Lemma 21.** *Let $A = \sum_{i=1}^{k} \eta_i^2 \mathbf{u}_i \mathbf{u}_i^t$ be a positive semidefinite symmetric matrix for $k \leq d$. Let $\mathbf{u}_1, \mathbf{u}_2, \ldots \mathbf{u}_k$ span a $k-1$ dimensional space. Let $B = A + R$, where $\|R\| \leq \epsilon$. Let $\mathbf{v}_1, \mathbf{v}_2, \ldots \mathbf{v}_{k-1}$ be the top $k-1$ eigenvectors of $B$. Then the projection of $\mathbf{u}_i$ in space orthogonal to $\mathbf{v}_1, \mathbf{v}_2, \ldots \mathbf{v}_{k-1}$ is $\leq \frac{2\sqrt{\epsilon}}{\eta_i}$.*

*Proof.* Let $\lambda_i^B$ be the $i^{th}$ largest eigenvalue of $B$. Observe that $B + \epsilon \mathbb{I}_d$ is a positive semidefinite matrix as for any vector $\mathbf{v}$, $\mathbf{v}^t (A + R + \epsilon \mathbb{I}_d) \mathbf{v} \geq 0$. Furthermore $\|A + R + \epsilon \mathbb{I}_d - A\| \leq 2\epsilon$. Since eigenvalues of $B + \epsilon \mathbb{I}_d$ is $\lambda^B + \epsilon$, by Lemma 20, for all $i \leq d$, $|\lambda_i^A - \lambda_i^B - \epsilon| \leq 2\epsilon$. Therefore, $|\lambda_i^B|$ for $i \geq k$ is $\leq 3\epsilon$.

Let $\mathbf{u}_i = \sum_{j=1}^{k-1} \alpha_{i,j} \mathbf{v}_j + \sqrt{1 - \sum_{j=1}^{k-1} \alpha_{i,j}^2} \mathbf{u}'$, for a vector $\mathbf{u}'$ orthogonal to $\mathbf{v}_1, \mathbf{v}_2, \ldots \mathbf{v}_{k-1}$. We compute $\mathbf{u}'^t A \mathbf{u}'$ in two ways. Since $A = B - R$,

$$|\mathbf{u}'^t (B-R) \mathbf{u}'| \leq |\mathbf{u}'^t B \mathbf{u}'| + |\mathbf{u}'^t R \mathbf{u}'| \leq \|B \mathbf{u}'\|_2 + \|R\|.$$

Since $\mathbf{u}'$ is orthogonal to first $k$ eigenvectors, we have $\|B\mathbf{u}'\|_2 \leq 3\epsilon$ and hence $|\mathbf{u}'^( B - R) \mathbf{u}'| \leq 4\epsilon$.

$$\mathbf{u}'^t A \mathbf{u}' \geq \eta_i^2 \Big( 1 - \sum_{j=1}^{k-1} \alpha_{i,j}^2 \Big).$$

We have shown that the above quantity is $\leq 4\epsilon$. Therefore $\big( 1 - \sum_{j=1}^{k-1} \alpha_{i,j}^2 \big)^{1/2} \leq 2\sqrt{\epsilon}/\eta_i$. $\qquad \square$

# B   Selection from a set of candidate distributions

Given samples from an unknown distribution $f$, the objective is to output a distribution from a known collection $\mathcal{F}$ of distributions with $\ell_1$ distance close to $D(f, \mathcal{F})$. Scheffe estimate [13] outputs a distribution from $\mathcal{F}$ whose $\ell_1$ distance from $f$ is at most $9.1 \max(D(f, \mathcal{F}), \epsilon)$ The algorithm requires $\mathcal{O}(\epsilon^{-2} \log |\mathcal{F}|)$ samples and the runs in time $\mathcal{O}(|\mathcal{F}|^2 T(n + |\mathcal{X}|))$, where $T$ is the time to compute the probability $f_j(x)$ of $x$, for any $f_j \in \mathcal{F}$. We present the modified Scheffe algorithm with near linear time complexity and then prove Lemma 1.

We first present the algorithm SCHEFFE* with running time $\widetilde{\mathcal{O}}(|\mathcal{F}|^2 T n)$.

---

Algorithm SCHEFFE*
**Input:** a set $\mathcal{F}$ of candidate distributions, $\epsilon$ : upper bound on $D(f, \mathcal{F})$, $n$ independent samples $x_1, \ldots, x_n$ from $f$.

For each pair $(p, q)$ in $\mathcal{F}$ do:

    1. $\mu_f = \frac{1}{n} \sum_{i=1}^{n} \mathbb{I}\{p(x_i) > q.(x_i)\}$.

    2. Generate independent samples $y_1, \ldots, y_n$ and $z_1, \ldots, z_n$ from $p$ and $q$ respectively.

    3. $\mu_p = \frac{1}{n} \sum_{i=1}^{n} \mathbb{I}\{p(y_i) > q(y_i)\}$, $\mu_q = \frac{1}{n} \sum_{i=1}^{n} \mathbb{I}\{p(z_i) > q(z_i)\}$.

    4. If $|\mu_p - \mu_f| < |\mu_q - \mu_f|$ declare $p$ as winner, else $q$.

Output the distribution with most wins, breaking ties arbitrarily.

---

We make the following modification to the algorithm where we reduce the size of potential distributions by half in every iteration.

---

Algorithm MODIFIED SCHEFFE

**Input:** set $\mathcal{F}$ of candidate distributions, $\epsilon$ : upper bound on $\min_{f_i \in \mathcal{F}} D(f, f_i)$, $n$ independent samples $x_1, \ldots, x_n$ from $f$.

1. Let $\mathcal{G} = \mathcal{F}$, $\mathcal{C} \leftarrow \varnothing$

2. Repeat until $|\mathcal{G}| > 1$:

    (a) Randomly form $|\mathcal{G}|/2$ pairs of distributions in $\mathcal{G}$ and run SCHEFFE* on *each pair* using the $n$ samples.
    (b) Replace $\mathcal{G}$ with the $|\mathcal{G}|/2$ winners.
    (c) Randomly select a set $\mathcal{A}$ of $\min\{|\mathcal{G}|, |\mathcal{F}|^{1/3}\}$ elements from $\mathcal{G}$.
    (d) Run SCHEFFE* on each pair in $\mathcal{A}$ and add the distributions with most wins to $\mathcal{C}$.

3. Run SCHEFFE* on $\mathcal{C}$ and output the winner

---

**Remark 22.** *For the ease of proof, we assume that $\delta \geq \frac{10 \log |\mathcal{F}|}{|\mathcal{F}|^{1/3}}$. If $\delta < \frac{10 \log |\mathcal{F}|}{|\mathcal{F}|^{1/3}}$, we run the algorithm with error probability $1/3$ and repeat it $\mathcal{O}(\log \frac{1}{\delta})$ times to choose a set of candidate mixtures $\mathcal{F}_\delta$. By Chernoff-bound with error probability $\leq \delta$, $\mathcal{F}_\delta$ contains a mixture close to $f$. Finally, we run SCHEFFE* on $\mathcal{F}_\delta$ to obtain a mixture that is close to $f$.*

*Proof sketch of Lemma 1.* For any set $\mathcal{A}$ and a distribution $p$, given $n$ independent samples from $p$ the empirical probability $\mu_n(\mathcal{A})$ has a distribution around $p(\mathcal{A})$ with standard deviation $\sim \frac{1}{\sqrt{n}}$. Together with an observation in Scheffe estimation in [13] one can show that if the number of samples $n = \mathcal{O}\left(\frac{\log \frac{|\mathcal{F}|}{\delta}}{\epsilon^2}\right)$, then SCHEFFE* has a guarantee $10 \max(\epsilon, D(f, \mathcal{F}))$ with probability $\geq 1 - \delta$.

Since we run SCHEFFE* at most $|\mathcal{F}|(2 \log |\mathcal{F}| + 1)$ times, choosing $\delta = \delta/(4|\mathcal{F}| \log |\mathcal{F}| + 2|\mathcal{F}|)$ results in the sample complexity of

$$\mathcal{O}\left(\frac{\log \frac{|\mathcal{F}|^2 (4 \log |\mathcal{F}| + 2)}{\delta}}{\epsilon^2}\right) = \mathcal{O}\left(\frac{\log \frac{|\mathcal{F}|}{\delta}}{\epsilon^2}\right),$$

and the total error probability of $\delta/2$ for all runs of SCHEFFE* during the algorithm. The above value of $n$ dictates our sample complexity. We now consider the following two cases:

- If at some stage $\geq \frac{\log(2/\delta)}{|\mathcal{F}|^{1/3}}$ fraction of elements in $\mathcal{A}$ have an $\ell_1$ distance $\leq 10\epsilon$ from $f$, then at that stage with probability $\geq 1 - \delta/2$ an element with distance $\leq 10\epsilon$ from $f$ is added to $\mathcal{A}$. Therefore a distribution with distance $\leq 100\epsilon$ is selected to $\mathcal{C}$.

- If at no stage this happens, then consider the element that is closest to $f$, *i.e.,* at $\ell_1$ distance at most $\epsilon$. With probability $\geq \left(1 - \frac{\log(2/\delta)}{|\mathcal{F}|^{1/3}}\right)^{\log |\mathcal{F}|}$ it always competes with an element at a distance at least $10\epsilon$ from $f$ and it wins all these games with probability $\geq 1 - \delta/2$.

Therefore with probability $\geq 1 - \delta/2$ there is an element in $\mathcal{C}$ at $\ell_1$ distance at most $100\epsilon$. Running SCHEFFE* on this set yields a distribution at a distance $\leq 100 \cdot 10\epsilon = 1000\epsilon$. The error probability is $\leq \delta$ by the union bound. $\qquad \square$

## C   Lower bound

We first show a lower bound for a single Gaussian distribution and generalize it to mixtures.

### C.1   Single Gaussian distribution

The proof is an application of the following version of Fano's inequality. It states that we cannot simultaneously estimate all distributions in a class using $n$ samples if they satisfy certain conditions.

**Lemma 23.** *[32] Let $f_1, \ldots, f_{r+1}$ be a collection of distributions such that for any $i \neq j$, $D(f_i, f_j) \geq \alpha$, and $KL(f_i, f_j) \leq \beta$. Let $f$ be an estimate of the underlying distribution using $n$ i.i.d. samples from one of the $f_i$'s. Then,*

$$\sup_i \mathbb{E}[D(f_i, f)] \geq \frac{\alpha}{2}\Big(1 - \frac{n\beta + \log 2}{\log r}\Big).$$

We consider $d-$dimensional spherical Gaussians with identity covariance matrix, with means along any coordinate restricted to $\pm\frac{c\epsilon}{\sqrt{d}}$. The KL divergence between two spherical Gaussians with identity covariance matrix is the squared distance between their means. Therefore, any two distributions we consider have KL distance at most

$$\beta = \sum_{i=1}^{d}\Big(2\frac{c\epsilon}{\sqrt{d}}\Big)^2 = 4c^2\epsilon^2,$$

We now consider a subset of these $2^d$ distributions to obtain a lower bound on $\alpha$. By the Gilbert-Varshamov bound, there exists a binary code with $\geq 2^{d/8}$ codewords of length $d$ and minimum distance $d/8$. Consider one such code. Now for each codeword, map $1 \to \frac{c\epsilon}{\sqrt{d}}$ and $0 \to -\frac{c\epsilon}{\sqrt{d}}$ to obtain a distribution in our class. We consider this subset of $\geq 2^{d/8}$ distributions as our $f_i$'s.

Consider any two $f_i$'s. Their means differ in at least $d/8$ coordinates. We show that the $\ell_1$ distance between them is $\geq c\epsilon/4$. Without loss of generality, let the means differ in the first $d/8$ coordinates, and furthermore, one of the distributions has means $c\epsilon/\sqrt{d}$ and the other has $-c\epsilon/\sqrt{d}$ in the first $d/8$ coordinates. The sum of the first $d/8$ coordinates is $N(c\epsilon\sqrt{d}/8, d/8)$ and $N(-c\epsilon\sqrt{d}/8, d/8)$. The $\ell_1$ distance between these normal random variables is a lower bound on the $\ell_1$ distance of the original random variables. For small values of $c\epsilon$ the distance between the two Gaussians is at least $\geq c\epsilon/4$. This serves as our $\alpha$.

Applying the Fano's Inequality, the $\ell_1$ error on the worst distribution is at least

$$\frac{c\epsilon}{8}\Big(1 - \frac{n4c^2\epsilon^2 + \log 2}{d/8}\Big),$$

which for $c = 16$ and $n < \frac{d}{2^{14}\epsilon^2}$ is at least $\epsilon$. In other words, the smallest $n$ to approximate all spherical normal distributions to $\ell_1$ distance at most $\epsilon$ is $> \frac{d}{2^{14}\epsilon^2}$.

## C.2  Mixtures of $k$ Gaussians

We now provide a lower bound on the sample complexity of learning mixtures of $k$ Gaussians in $d$ dimensions. We extend the construction for learning a single spherical Gaussian to mixtures of $k$ Gaussians and show a lower bound of $\Omega(kd/\epsilon^2)$ samples. We will again use Fano's inequality over a class of $2^{kd/64}$ distributions as described next.

To prove the lower bound on the sample complexity of learning spherical Gaussians, we designed a class of $2^{d/8}$ distributions around the origin. Let $\mathcal{P} \stackrel{\text{def}}{=} \{P_1, \ldots, P_T\}$, where $T = 2^{d/8}$, be this class. Recall that each $P_i$ is a spherical Gaussian with unit variance. For a distribution $P$ over $\mathbb{R}^d$ and $\boldsymbol{\mu} \in \mathbb{R}^d$, let $P + \boldsymbol{\mu}$ be the distribution $P$ shifted by $\boldsymbol{\mu}$.

We now choose $\boldsymbol{\mu}_1, \ldots, \boldsymbol{\mu}_k$'s *extremely well-separated*. The class of distributions we consider will be a mixture of $k$ components, where the $j$th component is a distribution from $\mathcal{P}$ shifted by $\boldsymbol{\mu}_j$. Since the $\boldsymbol{\mu}$'s will be well separated, we will use the results from last section over each component.

For $i \in [T]$, and $j \in [k]$, $P_{ij} \stackrel{\text{def}}{=} P_i + \boldsymbol{\mu}_j$. Each $(i_1, \ldots, i_k) \in [T]^k$ corresponds to the mixture

$$\frac{1}{k}(P_{i_1 1} + P_{i_2 2} + \ldots + P_{i_k k})$$

of $k$ spherical Gaussians. We consider this class of $T^k = 2^{kd/8}$ distributions. By the Gilbert-Varshamov bound, for any $T \geq 2$, there is a $T$-ary codes of length $k$, with minimum distance $\geq k/8$ and number of codewords $\geq 2^{k/8}$. This implies that among the $T^k = 2^{dk/8}$ distributions, there are

$2^{kd/64}$ distributions such that any two tuples $(i_1, \ldots, i_k)$ and $(i'_1, \ldots, i'_k)$ corresponding to different distributions differ in at least $k/8$ locations.

If we choose the $\boldsymbol{\mu}$'s well separated, the components of any mixture distribution have very little overlap. For simplicity, we choose $\boldsymbol{\mu}_j$'s satisfying

$$\min_{j_1 \neq j_2} \|\boldsymbol{\mu}_{j_1} - \boldsymbol{\mu}_{j_2}\|_2 \geq \left(\frac{2kd}{\epsilon}\right)^{100}.$$

This implies that for $j \neq l$, $\|P_{ij} - P_{i'l}\|_1 < (\epsilon/2dk)^{10}$. Therefore, for two different mixture distributions,

$$\left\|\frac{1}{k}(P_{i_1 1} + P_{i_2 2} + \ldots + P_{i_k k}) - \frac{1}{k}(P_{i'_1 1} + P_{i'_2 2} + \ldots + P_{i'_k k})\right\|_1$$

$$\overset{(a)}{\geq} \frac{1}{k} \sum_{j \in [k], i_j, i'_j \in [T]} |P_{i_j j} - P_{i'_j j}| - k^2(\epsilon/2dk)^{10}$$

$$\overset{(b)}{\geq} \frac{1}{8}\frac{c\epsilon}{4} - k^2(\epsilon/2dk)^{10}.$$

where $(a)$ follows form the fact that two mixtures have overlap only in the corresponding components, $(b)$ uses the fact that at least in $k/8$ components $i_j \neq i'_j$, and then uses the lower bound from the previous section.

Therefore, the $\ell_1$ distance between any two of the $2^{kd/64}$ distributions is $\geq c_1\epsilon/32$ for $c_1$ slightly smaller than $c$. We take this as $\alpha$.

Now, to upper bound the KL divergence, we simply use the convexity, namely for any distributions $P_1 \ldots P_k$ and $Q_1 \ldots Q_k$, let $\bar{P}$ and $\bar{Q}$ be the mean distributions. Then,

$$D(\bar{P}\|\bar{Q}) \leq \frac{1}{k}\sum_{i=1}^{k} D(P_i\|Q_i).$$

By the construction and from the previous section, for any $j$,

$$D(P_{i_j j}\|P_{i'_j j}) = D(P_i\|P_{i'}) \leq 4c^2\epsilon^2.$$

Therefore, we can take $\beta = 4c^2\epsilon^2$.

Therefore by the Fano's inequality, the $\ell_1$ error on the worst distribution is at least

$$\frac{c_1\epsilon}{64}\left(1 - \frac{n4c^2\epsilon^2 + \log 2}{dk/64}\right),$$

which for $c_1 = 128, c = 128.1$ and $n < \frac{dk}{8^8\epsilon^2}$ is at least $\epsilon$.

## D   One dimensional mixtures

### D.1   Proof of Lemma 3

The density of $N(\mu, \sigma^2)$ is $\geq (7\sigma)^{-1}$ in the interval $[\mu - \sqrt{2}\sigma, \mu + \sqrt{2}\sigma]$. Therefore, the probability that a sample occurs in the interval $\mu - \epsilon\sigma, \mu + \epsilon\sigma$ is $\geq 2\epsilon/7$. Hence, the probability that none of the $n$ samples occurs in $[\mu - \epsilon\sigma, \mu + \epsilon\sigma]$ is $\leq (1 - 2\epsilon/7)^n \leq e^{-2n\epsilon/7}$. If $\epsilon \geq \frac{7\log 2/\delta}{2n}$, then the probability that none of the samples occur in the interval is $\leq \delta/2$. A similar argument shows that there is a sample within interval, $[\mu + \sigma - \epsilon\sigma, \mu + \sigma + \epsilon\sigma]$, proving the lemma.

### D.2   Proof fo Lemma 4

Let $f = (w_1, w_2, \ldots w_k, p_1, p_2, \ldots p_k)$. For $\hat{f} = (\hat{w}_1, \hat{w}_2, \ldots, \hat{w}_{k-1}, 1 - \sum_{i=1}^{k-1}\hat{w}_i, \hat{p}_1, \hat{p}_2, \ldots \hat{p}_k)$, by the triangle inequality,

$$D(f, \hat{f}) \leq \sum_{i=1}^{k-1} 2|\hat{w}_i - w_i| + \sum_{i=1}^{k} w_i D(p_i, \hat{p}_i).$$

We show that there is a distribution in $\hat{f} \in \mathcal{F}$ such that the sum above is bounded by $\epsilon$. Since we quantize the grids as multiples of $\epsilon/2k$, we consider distributions in $\mathcal{F}$ such that each $|\hat{w}_i - w_i| \leq \epsilon/4k$, and therefore $\sum_i |\hat{w}_i - w_i| \leq \frac{\epsilon}{2}$.

We now show that for each $p_i$ there is a $\hat{p}_i$ such that $w_i D(p_i, \hat{p}_i) \leq \frac{\epsilon}{2k}$, thus proving that $D(f, \hat{f}) \leq \epsilon$. If $w_i \leq \frac{\epsilon}{4k}$, then $w_i D(p_i, \hat{p}_i) \leq \frac{\epsilon}{2k}$. Otherwise, let $w_i' > \frac{\epsilon}{4k}$ be the fraction of samples from $p_i$. By Lemma 3 and 14, with probability $\geq 1 - \delta/2k$,

$$D(p_i, \hat{p}_i)^2 \leq 2\frac{(\mu_i - \mu_i')^2}{\sigma_i^2} + 16\frac{(\sigma_i - \sigma_i')^2}{\sigma_i^2}$$

$$\leq \frac{25 \log^2 \frac{4k}{\delta}}{(nw_i')^2} + \frac{800 \log^2 \frac{4k}{\delta}}{(nw_i')^2}$$

$$\leq \frac{825 \log^2 \frac{4k}{\delta}}{(nw_i')^2}.$$

Therefore,

$$w_i D(p_i, \hat{p}_i) \leq \frac{30 w_i \log \frac{4k}{\delta}}{nw_i'}.$$

Since $w_i > \epsilon/4k$, with probability $\geq 1 - \delta/2k$, $w_i \leq 2w_i'$. By the union bound with probability $\geq 1 - \delta/k$, $w_i D(p_i, \hat{p}_i) \leq \frac{60 \log \frac{4k}{\delta}}{n}$. Hence if $n \geq \frac{120k \log \frac{4k}{\delta}}{\epsilon}$, the above quantity is less than $\epsilon/2k$. The total error probability is $\leq \delta$ by the union bound.

### D.3 Proof of Corollary 5

Use $n' \stackrel{\text{def}}{=} \frac{120k \log \frac{4k}{\delta}}{\epsilon}$ samples to generate a set of at most $n'^{3k-1}$ candidate distributions as stated in Lemma 4. With probability $\geq 1 - \delta$, one of the candidate distributions is $\epsilon$-close to the underlying one. Run MODIFIED SCHEFFE on this set of candidate distributions to obtain a $1000\epsilon$-close estimate of $f$ with probability $\geq 1 - \delta$ (Lemma 1). The run time is dominated by the run time of MODIFIED SCHEFFE which is $\mathcal{O}\left(\frac{|\mathcal{F}|T \log \frac{|\mathcal{F}|}{\delta}}{\epsilon^2}\right)$, where $|\mathcal{F}| = n'^{3k-1}$ and $T = k$. The total error probability is $\leq 2\delta$ by the union bound.

## E  Proofs for $k$ spherical Gaussians

We first state a simple concentration result that helps us in other proofs.

**Lemma 24.** *Given $n$ samples from a set of Gaussian distributions, with probability $\geq 1 - 2\delta$, for every pair of samples $\mathbf{X} \sim N(\boldsymbol{\mu}_1, \sigma^2 \mathbb{I}_d)$ and $\mathbf{Y} \sim N(\boldsymbol{\mu}_2, \sigma^2 \mathbb{I}_d)$,*

$$\|\mathbf{X} - \mathbf{Y}\|_2^2 \leq 2d\sigma^2 + 4\sigma^2\sqrt{d \log \frac{n^2}{\delta}} + \|\boldsymbol{\mu}_1 - \boldsymbol{\mu}_2\|_2^2 + 4\sigma \|\boldsymbol{\mu}_1 - \boldsymbol{\mu}_2\|_2 \sqrt{\log \frac{n^2}{\delta}} + 4\sigma^2 \log \frac{n^2}{\delta}. \quad (1)$$

*and*

$$\|\mathbf{X} - \mathbf{Y}\|_2^2 \geq 2d\sigma^2 - 4\sigma^2\sqrt{d \log \frac{n^2}{\delta}} + \|\boldsymbol{\mu}_1 - \boldsymbol{\mu}_2\|_2^2 - 4\sigma \|\boldsymbol{\mu}_1 - \boldsymbol{\mu}_2\|_2 \sqrt{\log \frac{n^2}{\delta}}. \quad (2)$$

*Proof.* We prove the lower bound, the proof for the upper bound is similar and omitted. Since $\mathbf{X}$ and $\mathbf{Y}$ are Gaussians, $\mathbf{X} - \mathbf{Y}$ is distributed as $N(\boldsymbol{\mu}_1 - \boldsymbol{\mu}_2, 2\sigma^2)$. Rewriting $\|\mathbf{X} - \mathbf{Y}\|_2$

$$\|\mathbf{X} - \mathbf{Y}\|_2^2 = \|\mathbf{X} - \mathbf{Y} - (\boldsymbol{\mu}_1 - \boldsymbol{\mu}_2)\|_2^2 + \|\boldsymbol{\mu}_1 - \boldsymbol{\mu}_2\|_2^2 + 2(\boldsymbol{\mu}_1 - \boldsymbol{\mu}_2) \cdot (\mathbf{X} - \mathbf{Y} - (\boldsymbol{\mu}_1 - \boldsymbol{\mu}_2)).$$

Let $\mathbf{Z} = \mathbf{X} - \mathbf{Y} - (\boldsymbol{\mu}_1 - \boldsymbol{\mu}_2)$, then $\mathbf{Z} \sim N(\mathbf{0}, 2\sigma^2 \mathbb{I}_d)$. Therefore by Lemma 16, with probability $1 - \delta/n^2$,

$$\|\mathbf{Z}\|_2^2 \geq 2d\sigma^2 - 4\sigma^2\sqrt{d \log \frac{n^2}{\delta}}.$$

Furthermore $(\boldsymbol{\mu}_1 - \boldsymbol{\mu}_2) \cdot \mathbf{Z}$ is sum of Gaussians and hence a Gaussian distribution. It has mean $0$ and variance $2\sigma^2 \|\boldsymbol{\mu}_1 - \boldsymbol{\mu}_2\|_2^2$. Therefore, by Lemma 15 with probability $1 - \delta/n^2$,

$$(\boldsymbol{\mu}_1 - \boldsymbol{\mu}_2) \cdot \mathbf{Z} \geq -2\sigma \|\boldsymbol{\mu}_1 - \boldsymbol{\mu}_2\|_2 \sqrt{\log \frac{n^2}{\delta}}.$$

By the union bound with probability $1 - 2\delta/n^2$,

$$\|\mathbf{X} - \mathbf{Y}\|_2^2 \geq 2d\sigma^2 - 4\sigma^2 \sqrt{d \log \frac{n^2}{\delta}} + \|\boldsymbol{\mu}_1 - \boldsymbol{\mu}_2\|_2^2 - 4\sigma \|\boldsymbol{\mu}_1 - \boldsymbol{\mu}_2\|_2 \sqrt{\log \frac{n^2}{\delta}}.$$

There are $\binom{n}{2}$ pairs and the lemma follows by the union bound. $\qquad\square$

## E.1 Proof of Lemma 7

We show that if Equations (1) and (2) are satisfied, then the lemma holds. The error probability is that of Lemma 24 and is $\leq 2\delta$. Since the minimum is over $k + 1$ indices, at least two samples are from the same component. Applying Equations (1) and (2) for these two samples

$$2d\hat{\sigma}^2 \leq 2d\sigma^2 + 4\sigma^2 \sqrt{d \log \frac{n^2}{\delta}} + 4\sigma^2 \log \frac{n^2}{\delta}.$$

Similarly by Equations (1) and (2) for any two samples $\mathbf{X}(a), \mathbf{X}(b)$ in $[k + 1]$,

$$\|\mathbf{X}(a) - \mathbf{X}(b)\|_2^2 \geq 2d\sigma^2 - 4\sigma^2 \sqrt{d \log \frac{n^2}{\delta}} + \|\boldsymbol{\mu}_i - \boldsymbol{\mu}_j\|_2^2 - 4\sigma \|\boldsymbol{\mu}_i - \boldsymbol{\mu}_j\|_2 \sqrt{\log \frac{n^2}{\delta}}$$

$$\geq 2d\sigma^2 - 4\sigma^2 \sqrt{d \log \frac{n^2}{\delta}} - 4\sigma^2 \log \frac{n^2}{\delta},$$

where the last inequality follows from the fact that $\alpha^2 - 4\alpha\beta \geq -4\beta^2$. The result follows from the assumption that $d > 20 \log n^2/\delta$.

## E.2 Proof of Lemma 8

We show that if Equations (1) and (2) are satisfied, then the lemma holds. The error probability is that of Lemma 24 and is $\leq 2\delta$. Since Equations (1) and (2) are satisfied, by the proof of Lemma 7, $|\hat{\sigma}^2 - \sigma^2| \leq 2.5\sigma^2 \sqrt{\frac{\log(n^2/\delta)}{d}}$. If two samples $X(a)$ and $X(b)$ are from the same component, by Lemma 24,

$$\|\mathbf{X}(a) - \mathbf{X}(b)\|_2^2 \leq 2d\sigma^2 + 4\sigma^2 \sqrt{d \log \frac{n^2}{\delta}} + 4\sigma^2 \log \frac{n^2}{\delta} \leq 2d\sigma^2 + 5\sigma^2 \sqrt{d \log \frac{n^2}{\delta}}.$$

By Lemma 7, the above quantity is less than $2d\hat{\sigma}^2 + 23\hat{\sigma}^2 \sqrt{d \log \frac{n^2}{\delta}}$. Hence all the samples from the same component are in a single cluster.

Suppose there are two samples from different components in a cluster, then by Equations (1) and (2),

$$2d\hat{\sigma}^2 + 23\hat{\sigma}^2 \sqrt{d \log \frac{n^2}{\delta}} \geq 2d\sigma^2 - 4\sigma^2 \sqrt{d \log \frac{n^2}{\delta}} + \|\boldsymbol{\mu}_i - \boldsymbol{\mu}_j\|_2^2 - 4\sigma \|\boldsymbol{\mu}_i - \boldsymbol{\mu}_j\|_2 \sqrt{\log \frac{n^2}{\delta}}.$$

Relating $\hat{\sigma}^2$ and $\sigma^2$ using Lemma 7,

$$2d\sigma^2 + 40\sigma^2 \sqrt{d \log \frac{n^2}{\delta}} \geq 2d\sigma^2 - 4\sigma^2 \sqrt{d \log \frac{n^2}{\delta}} + \|\boldsymbol{\mu}_i - \boldsymbol{\mu}_j\|_2^2 - 4\sigma \|\boldsymbol{\mu}_i - \boldsymbol{\mu}_j\|_2 \sqrt{\log \frac{n^2}{\delta}}.$$

Hence $\|\boldsymbol{\mu}_i - \boldsymbol{\mu}_j\|_2 \leq 10\sigma \left(d \log \frac{n^2}{\delta}\right)^{1/4}$. There are at most $k$ components; therefore, any two components within the same cluster are at a distance $\leq 10k\sigma \left(d \log \frac{n^2}{\delta}\right)^{1/4}$.

### E.3 Proof of Lemma 9

The proof is involved and we show it in steps. We first show few concentration bounds which we use later to argue that the samples are clusterable when the sample covariance matrix has a large eigenvalue. Let $\hat{w}_i$ be the fraction of samples from component $i$. Let $\hat{\mu}_i$ be the empirical average of samples from $\mathbf{p}_i$. Let $\hat{\bar{\mu}}(C)$ be the empirical average of samples in cluster $C$. If $C$ is the entire set of samples we use $\hat{\bar{\mu}}$ instead of $\hat{\bar{\mu}}(C)$. We first show a concentration inequality that we use in rest of the calculations.

**Lemma 25.** *Given $n$ samples from a $k$-component Gaussian mixture with probability $\geq 1 - 2\delta$, for every component $i$*

$$\|\hat{\mu}_i - \mu_i\|_2^2 \leq \left(d + 3\sqrt{d\log\frac{2k}{\delta}}\right)\frac{\sigma^2}{n\hat{w}_i} \text{ and } |\hat{w}_i - w_i| \leq \sqrt{\frac{2w_i\log\frac{2k}{\delta}}{n}} + \frac{2}{3}\frac{\log\frac{2k}{\delta}}{n}. \tag{3}$$

*Proof.* Since $\hat{\mu}_i - \mu_i$ is distributed $N(0, \sigma^2\mathbb{I}_d/n\hat{w}_i)$, by Lemma 16 with probability $\geq 1 - \delta/k$,

$$\|\hat{\mu}_i - \mu_i\|_2^2 \leq \left(d + 2\sqrt{d\log\frac{2k}{\delta}} + 2\log\frac{2k}{\delta}\right)\frac{\sigma^2}{n\hat{w}_i} \leq \left(d + 3\sqrt{d\log\frac{2k}{\delta}}\right)\frac{\sigma^2}{n\hat{w}_i}.$$

The second inequality uses the fact that $d \geq 20\log n^2/\delta$. For bounding the weights, observe that by Lemma 18 with probability $\geq 1 - \delta/k$,

$$|\hat{w}_i - w_i| \leq \sqrt{\frac{2w_i\log 2k/\delta}{n}} + \frac{2}{3}\frac{\log 2k/\delta}{n}.$$

By the union bound the error probability is $\leq 2k\delta/2k = \delta$. $\qquad\square$

A simple application of triangle inequality yields the following lemma.

**Lemma 26.** *Given $n$ samples from a $k$-component Gaussian mixture if Equation (3) holds, then*

$$\left\|\sum_{i=1}^k \hat{w}_i(\hat{\mu}_i - \mu_i)(\hat{\mu}_i - \mu_i)^t\right\| \leq \left(d + 3\sqrt{d\log\frac{2k}{\delta}}\right)\frac{k\sigma^2}{n}.$$

**Lemma 27.** *Given $n$ samples from a $k$-component Gaussian mixture, if Equation (3) holds and the maximum distance between two components is $\leq 10k\sigma\left(d\log\frac{n^2}{\delta}\right)^{1/4}$, then $\left\|\hat{\bar{\mu}} - \bar{\mu})\right\|_2 \leq c\sigma\sqrt{\frac{dk\log\frac{n^2}{\delta}}{n}}$, for a constant $c$.*

*Proof.* Observe that

$$\hat{\bar{\mu}} - \bar{\mu} = \sum_{i=1}^k \hat{w}_i\hat{\mu}_i - w_i\mu_i = \sum_{i=1}^k \hat{w}_i(\hat{\mu}_i - \mu_i) + (\hat{w}_i - w_i)\mu_i = \sum_{i=1}^k \hat{w}_i(\hat{\mu}_i - \mu_i) + (\hat{w}_i - w_i)(\mu_i - \bar{\mu}). \tag{4}$$

Hence by Equation (3) and the fact that the maximum distance between two components is $\leq 10k\sigma\left(d\log\frac{n^2}{\delta}\right)^{1/4}$,

$$\left\|\hat{\bar{\mu}} - \bar{\mu}\right\|_2 \leq \sum_{i=1}^k \hat{w}_i\sqrt{\left(d + 3\sqrt{d\log\frac{2k}{\delta}}\right)}\frac{\sigma}{\sqrt{n\hat{w}_i}} + \left(\sqrt{\frac{2w_i\log 2k/\delta}{n}} + \frac{2}{3}\frac{\log 2k/\delta}{n}\right)10k\left(d\log\frac{n^2}{\delta}\right)^{1/4}\sigma.$$

For $n \geq d \geq \max(k^4, 20\log n^2/\delta, 1000)$, we get the above term is $\leq c\sqrt{\frac{kd\log n^2/\delta}{n}}\sigma$, for some constant $c$. $\qquad\square$

We now make a simple observation on covariance matrices.

**Lemma 28.** *Given $n$ samples from a $k$-component mixture,*

$$\left\| \sum_{i=1}^{k} \hat{w}_i (\hat{\boldsymbol{\mu}}_i - \hat{\overline{\boldsymbol{\mu}}})(\hat{\boldsymbol{\mu}}_i - \hat{\overline{\boldsymbol{\mu}}})^t - \sum_{i=1}^{k} \hat{w}_i (\boldsymbol{\mu}_i - \overline{\boldsymbol{\mu}})(\boldsymbol{\mu}_i - \overline{\boldsymbol{\mu}})^t \right\|$$

$$\leq 2 \left\| \hat{\overline{\boldsymbol{\mu}}} - \overline{\boldsymbol{\mu}} \right\|_2^2 + \sum_{i=1}^{k} 2\hat{w}_i \left\| \hat{\boldsymbol{\mu}}_i - \boldsymbol{\mu}_i \right\|_2^2 + 2 \left( \sqrt{k} \left\| \hat{\overline{\boldsymbol{\mu}}} - \overline{\boldsymbol{\mu}} \right\|_2 + \sum_{i=1}^{k} \sqrt{\hat{w}_i} \left\| \hat{\boldsymbol{\mu}}_i - \boldsymbol{\mu}_i \right\|_2 \right) \max_j \sqrt{\hat{w}_j} \left\| \boldsymbol{\mu}_j - \overline{\boldsymbol{\mu}} \right\|_2.$$

*Proof.* Observe that for any two vectors $\mathbf{u}$ and $\mathbf{v}$,

$$\mathbf{u}\mathbf{u}^t - \mathbf{v}\mathbf{v}^t = \mathbf{u}(\mathbf{u}^t - \mathbf{v}^t) + (\mathbf{u} - \mathbf{v})\mathbf{v}^t = (\mathbf{u} - \mathbf{v})(\mathbf{u} - \mathbf{v})^t + \mathbf{v}(\mathbf{u} - \mathbf{v})^t + (\mathbf{u} - \mathbf{v})\mathbf{v}^t.$$

Hence by triangle inequality,

$$\left\| \mathbf{u}\mathbf{u}^t - \mathbf{v}\mathbf{v}^t \right\| \leq \left\| \mathbf{u} - \mathbf{v} \right\|_2^2 + 2 \left\| \mathbf{v} \right\|_2 \left\| \mathbf{u} - \mathbf{v} \right\|_2.$$

Applying the above observation to $\mathbf{u} = \hat{\boldsymbol{\mu}}_i - \hat{\overline{\boldsymbol{\mu}}}$ and $\mathbf{v} = \boldsymbol{\mu}_i - \overline{\boldsymbol{\mu}}$, we get

$$\sum_{i=1}^{k} \hat{w}_i \left\| (\hat{\boldsymbol{\mu}}_i - \hat{\overline{\boldsymbol{\mu}}})(\hat{\boldsymbol{\mu}}_i - \hat{\overline{\boldsymbol{\mu}}})^t - (\boldsymbol{\mu}_i - \overline{\boldsymbol{\mu}})(\boldsymbol{\mu}_i - \overline{\boldsymbol{\mu}})^t \right\|$$

$$\leq \sum_{i=1}^{k} \left( \hat{w}_i \left\| \hat{\boldsymbol{\mu}}_i - \hat{\overline{\boldsymbol{\mu}}} - \boldsymbol{\mu}_i - \overline{\boldsymbol{\mu}} \right\|_2^2 + 2\sqrt{\hat{w}_i} \left\| \boldsymbol{\mu}_i - \overline{\boldsymbol{\mu}} \right\|_2 \sqrt{\hat{w}_i} \left\| \hat{\boldsymbol{\mu}}_i - \hat{\overline{\boldsymbol{\mu}}} - \boldsymbol{\mu}_i - \overline{\boldsymbol{\mu}} \right\|_2 \right)$$

$$\leq \sum_{i=1}^{k} \left( 2\hat{w}_i \left\| \hat{\boldsymbol{\mu}}_i - \boldsymbol{\mu}_i \right\|_2^2 + 2\hat{w}_i \left\| \hat{\overline{\boldsymbol{\mu}}} - \overline{\boldsymbol{\mu}} \right\|_2^2 + 2 \max_j \sqrt{\hat{w}_j} \left\| \boldsymbol{\mu}_j - \overline{\boldsymbol{\mu}} \right\|_2 \left( \sqrt{\hat{w}_i} \left\| \hat{\boldsymbol{\mu}}_i - \boldsymbol{\mu}_i \right\|_2 + \sqrt{\hat{w}_i} \left\| \hat{\overline{\boldsymbol{\mu}}} - \overline{\boldsymbol{\mu}} \right\|_2 \right) \right)$$

$$\leq 2 \left\| \hat{\overline{\boldsymbol{\mu}}} - \overline{\boldsymbol{\mu}} \right\|_2^2 + \sum_{i=1}^{k} 2\hat{w}_i \left\| \hat{\boldsymbol{\mu}}_i - \boldsymbol{\mu}_i \right\|_2^2 + 2 \left( \sqrt{k} \left\| \hat{\overline{\boldsymbol{\mu}}} - \overline{\boldsymbol{\mu}} \right\|_2 + \sum_{i=1}^{k} \sqrt{\hat{w}_i} \left\| \hat{\boldsymbol{\mu}}_i - \boldsymbol{\mu}_i \right\|_2 \right) \max_j \sqrt{\hat{w}_j} \left\| \boldsymbol{\mu}_j - \overline{\boldsymbol{\mu}} \right\|_2.$$

The lemma follows from triangle inequality. $\qquad \square$

The following lemma immediately follows from Lemmas 27 and 28.

**Lemma 29.** *Given $n$ samples from a $k$-component Gaussian mixture, if Equation (3) and the maximum distance between two components is $\leq 10k\sigma \left( d \log \frac{n^2}{\delta} \right)^{1/4}$, then*

$$\left\| \sum_{i=1}^{k} \hat{w}_i (\hat{\boldsymbol{\mu}}_i - \hat{\overline{\boldsymbol{\mu}}})(\hat{\boldsymbol{\mu}}_i - \hat{\overline{\boldsymbol{\mu}}})^t - \sum_{i=1}^{k} \hat{w}_i (\boldsymbol{\mu}_i - \overline{\boldsymbol{\mu}})(\boldsymbol{\mu}_i - \overline{\boldsymbol{\mu}})^t \right\|$$

$$\leq \frac{c\sigma^2 dk^2 \log \frac{n^2}{\delta}}{n} + c\sigma \sqrt{\frac{dk^2 \log \frac{n^2}{\delta}}{n}} \max_i \sqrt{\hat{w}_i} \left\| \boldsymbol{\mu}_i - \overline{\boldsymbol{\mu}} \right\|_2,$$

*for a constant $c$.*

**Lemma 30.** *For a set of samples $\mathbf{X}(1), \dots \mathbf{X}(n)$ from a $k$-component mixture,*

$$\sum_{i=1}^{n} \frac{(\mathbf{X}(i) - \hat{\overline{\boldsymbol{\mu}}})(\mathbf{X}(i) - \hat{\overline{\boldsymbol{\mu}}})^t}{n} = \sum_{i=1}^{k} \hat{w}_i (\hat{\boldsymbol{\mu}}_i - \hat{\overline{\boldsymbol{\mu}}})(\hat{\boldsymbol{\mu}}_i - \hat{\overline{\boldsymbol{\mu}}})^t - \hat{w}_i (\hat{\boldsymbol{\mu}}_i - \boldsymbol{\mu}_i)(\hat{\boldsymbol{\mu}}_i - \boldsymbol{\mu}_i)^t$$

$$+ \sum_{j | \mathbf{X}(j) \sim p_i} \frac{(\mathbf{X}(j) - \boldsymbol{\mu}_i)(\mathbf{X}(j) - \boldsymbol{\mu}_i)^t}{n}.$$

*where $\hat{w}_i$ and $\hat{\boldsymbol{\mu}}_i$ are the empirical weights and averages of components $i$ and $\hat{\overline{\boldsymbol{\mu}}} = \frac{1}{n} \sum_{i=1}^{n} \mathbf{X}_i$.*

*Proof.* The given expression can be rewritten as

$$\frac{1}{n} \sum_{i=1}^{n} (\mathbf{X}(i) - \hat{\overline{\boldsymbol{\mu}}})(\mathbf{X}(i) - \hat{\overline{\boldsymbol{\mu}}})^t = \sum_{i=1}^{k} \hat{w}_i \sum_{j | \mathbf{X}(j) \sim p_i} \frac{1}{n\hat{w}_i} \mathbf{X}(j) - \hat{\overline{\boldsymbol{\mu}}}(\mathbf{X}(j) - \hat{\overline{\boldsymbol{\mu}}})^t.$$

First observe that for any set of points $x_i$ and their average $\hat{x}$ and any value $a$,

$$\sum_i (x_i - a)^2 = \sum_i (x_i - \hat{x})^2 + (\hat{x} - a)^2.$$

Hence for samples from a component $i$,

$$\sum_{j|\mathbf{X}(j)\sim p_i} \frac{1}{n\hat{w}_i}(\mathbf{X}(j)-\hat{\overline{\boldsymbol{\mu}}})(\mathbf{X}(j)-\hat{\overline{\boldsymbol{\mu}}})^t$$

$$= \sum_{j|\mathbf{X}(j)\sim p_i} \frac{1}{n\hat{w}_i}(\hat{\boldsymbol{\mu}}_i-\hat{\overline{\boldsymbol{\mu}}})(\hat{\boldsymbol{\mu}}_i-\hat{\overline{\boldsymbol{\mu}}})^t + \sum_{j|\mathbf{X}(j)\sim p_i} \frac{1}{n\hat{w}_i}(\mathbf{X}(j)-\hat{\boldsymbol{\mu}}_i)(\mathbf{X}(j)-\hat{\boldsymbol{\mu}}_i)^t$$

$$= (\hat{\boldsymbol{\mu}}_i-\hat{\overline{\boldsymbol{\mu}}})(\hat{\boldsymbol{\mu}}_i-\hat{\overline{\boldsymbol{\mu}}})^t + \sum_{j|\mathbf{X}(j)\sim p_i} \frac{1}{n\hat{w}_i}(\mathbf{X}(j)-\hat{\boldsymbol{\mu}}_i)(\mathbf{X}(j)-\hat{\boldsymbol{\mu}}_i)^t$$

$$= (\hat{\boldsymbol{\mu}}_i-\hat{\overline{\boldsymbol{\mu}}})(\hat{\boldsymbol{\mu}}_i-\hat{\overline{\boldsymbol{\mu}}})^t + \sum_{j|\mathbf{X}(j)\sim p_i} \frac{1}{n\hat{w}_i}(\mathbf{X}(j)-\boldsymbol{\mu}_i)(\mathbf{X}(j)-\boldsymbol{\mu}_i)^t - (\hat{\boldsymbol{\mu}}_i-\boldsymbol{\mu}_i)(\hat{\boldsymbol{\mu}}_i-\boldsymbol{\mu}_i)^t.$$

Summing over all components results in the lemma. □

We now bound the error in estimating the eigenvalue of the covariance matrix.

**Lemma 31.** *Given* $\mathbf{X}(1),\dots\mathbf{X}(n)$, $n$ *samples from a $k$-component Gaussian mixture, if Equations (1), (2), and (3) hold, then with probability* $\geq 1-2\delta$,

$$\left\| \frac{1}{n}\sum_{i=1}^{n}(\mathbf{X}(i)-\hat{\overline{\boldsymbol{\mu}}})(\mathbf{X}(i)-\hat{\overline{\boldsymbol{\mu}}})^t - \hat{\sigma}^2\mathbb{I}_d - \sum_{i=1}^{k}\hat{w}_i(\boldsymbol{\mu}_i-\overline{\boldsymbol{\mu}})(\boldsymbol{\mu}_i-\overline{\boldsymbol{\mu}})^t \right\|$$

$$\leq c(n) \stackrel{\text{def}}{=} c\sigma^2\sqrt{\frac{d\log\frac{n^2}{\delta}}{n}} + c\sigma^2\frac{dk^2\log\frac{n^2}{\delta}}{n} + c\sigma\sqrt{\frac{dk^2\log\frac{n^2}{\delta}}{n}}\max_i\sqrt{\hat{w}_i}\|\boldsymbol{\mu}_i-\overline{\boldsymbol{\mu}}\|_2 , \tag{5}$$

*for a constant $c$.*

*Proof.* Since Equations (1), (2), and (3) hold, conditions in Lemmas 27 and 29 are satisfied. By Lemma 29,

$$\left\| \sum_{i=1}^{k}\hat{w}_i(\hat{\boldsymbol{\mu}}_i-\hat{\overline{\boldsymbol{\mu}}})(\hat{\boldsymbol{\mu}}_i-\hat{\overline{\boldsymbol{\mu}}})^t - \sum_{i=1}^{k}\hat{w}_i(\boldsymbol{\mu}_i-\overline{\boldsymbol{\mu}})(\boldsymbol{\mu}_i-\overline{\boldsymbol{\mu}})^t \right\|$$

$$= \mathcal{O}\left( \sigma^2\frac{dk^2\log\frac{n^2}{\delta}}{n} + \sigma\sqrt{\frac{dk^2\log\frac{n^2}{\delta}}{n}}\max_i\sqrt{\hat{w}_i}\|\boldsymbol{\mu}_i-\overline{\boldsymbol{\mu}}\|_2 \right).$$

Hence it remains to show,

$$\left\| \frac{1}{n}\sum_{i=1}^{n}(\mathbf{X}(i)-\hat{\overline{\boldsymbol{\mu}}})(\mathbf{X}(i)-\hat{\overline{\boldsymbol{\mu}}})^t - \sum_{i=1}^{k}\hat{w}_i(\hat{\boldsymbol{\mu}}_i-\hat{\overline{\boldsymbol{\mu}}})(\hat{\boldsymbol{\mu}}_i-\hat{\overline{\boldsymbol{\mu}}})^t \right\| = \mathcal{O}\left( \sqrt{\frac{kd\log\frac{5k^2}{\delta}}{n}}\sigma^2 \right).$$

By Lemma 30, the covariance matrix can be rewritten as

$$\sum_{i=1}^{k}\hat{w}_i(\hat{\boldsymbol{\mu}}_i-\hat{\overline{\boldsymbol{\mu}}})(\hat{\boldsymbol{\mu}}_i-\hat{\overline{\boldsymbol{\mu}}})^t - \hat{w}_i(\hat{\boldsymbol{\mu}}_i-\boldsymbol{\mu}_i)(\hat{\boldsymbol{\mu}}_i-\boldsymbol{\mu}_i)^t + \sum_{i=1}^{k}\sum_{j|\mathbf{X}(j)\sim p_i}\frac{1}{n}(\mathbf{X}(j)-\boldsymbol{\mu}_i)(\mathbf{X}(j)-\boldsymbol{\mu}_i)^t - \hat{\sigma}^2\mathbb{I}_d. \tag{6}$$

We now bound the norms of second and third terms in the above equation. Consider the third term, $\sum_{i=1}^{k}\sum_{j|\mathbf{X}(j)\sim p_i}\frac{1}{n}(\mathbf{X}(j)-\boldsymbol{\mu}_i)(\mathbf{X}(j)-\boldsymbol{\mu}_i)^t$. Conditioned on the fact that $\mathbf{X}(j)\sim p_i$, $\mathbf{X}(j)-\boldsymbol{\mu}_i$ is distributed $N(0,\sigma^2\mathbb{I}_d)$, therefore by Lemma 19 and Lemma 7, with probability $\geq 1-2\delta$,

$$\left\| \sum_{i=1}^{k}\sum_{j|\mathbf{X}(j)\sim p_i}\frac{1}{n}(\mathbf{X}(j)-\boldsymbol{\mu}_i)(\mathbf{X}(j)-\boldsymbol{\mu}_i)^t - \hat{\sigma}^2\mathbb{I}_d \right\| \leq c'\sqrt{\frac{d\log\frac{2d}{\delta}}{n}}\sigma^2 + 2.5\sigma^2\sqrt{\frac{\log\frac{n^2}{\delta}}{d}}.$$

The second term in Equation (6) is bounded by Lemma 26. Hence together with the fact that $d \geq 20\log n^2/\delta$ we get that with probability $\geq 1-2\delta$, the second and third terms are bounded by $\mathcal{O}\left(\sigma^2\sqrt{\frac{dk}{n}\log\frac{n^2}{\delta}}\right)$. □

**Lemma 32.** *Let $\mathbf{u}$ be the largest eigenvector of the sample covariance matrix and $n \geq c \cdot dk^2 \log \frac{n^2}{\delta}$. If $\max_i \sqrt{\widehat{w}_i} \|\boldsymbol{\mu}_i - \overline{\boldsymbol{\mu}}\|_2 = \alpha\sigma$ and Equation (5) holds, then there exists $i$ such that $|\mathbf{u} \cdot (\boldsymbol{\mu}_i - \overline{\boldsymbol{\mu}})| \geq \sigma(\alpha - 1 - 1/\alpha)/\sqrt{k}$.*

*Proof.* Observe that $\left\|\sum_j w_j \mathbf{v}_j \mathbf{v}_j^t\right\| \geq \left\|\sum_j w_j \mathbf{v}_j \mathbf{v}_j^t \frac{\mathbf{v}_i}{\|\mathbf{v}_i\|}\right\|_2 \geq w_i \|\mathbf{v}_i\|_2^2$. Therefore

$$\left\|\sum_{i=1}^k \widehat{w}_i(\boldsymbol{\mu}_i - \overline{\boldsymbol{\mu}})(\boldsymbol{\mu}_i - \overline{\boldsymbol{\mu}})^t\right\| \geq \left\|\sum_{j=1}^k \widehat{w}_j(\boldsymbol{\mu}_j - \overline{\boldsymbol{\mu}})(\boldsymbol{\mu}_j - \overline{\boldsymbol{\mu}})^t(\boldsymbol{\mu}_i - \overline{\boldsymbol{\mu}})/\|\boldsymbol{\mu}_i - \overline{\boldsymbol{\mu}}\|\right\|_2 \geq \alpha^2\sigma^2.$$

Hence by Lemma 31 and the triangle inequality, the largest eigenvalue of the sample-covariance matrix is $\geq \alpha^2\sigma^2 - c(n)$. Similarly by applying Lemma 31 again we get, $\left\|\sum_{i=1}^k \widehat{w}_i(\boldsymbol{\mu}_i - \overline{\boldsymbol{\mu}})(\boldsymbol{\mu}_i - \overline{\boldsymbol{\mu}})^t\mathbf{u}\right\|_2 \geq \alpha^2\sigma^2 - 2c(n)$. By triangle inequality and Cauchy-Schwartz inequality,

$$\left\|\sum_{i=1}^k \widehat{w}_i(\boldsymbol{\mu}_i - \overline{\boldsymbol{\mu}})(\boldsymbol{\mu}_i - \overline{\boldsymbol{\mu}})^t\mathbf{u}\right\|_2 \leq \sum_{i=1}^k \left\|\widehat{w}_i(\boldsymbol{\mu}_i - \overline{\boldsymbol{\mu}})(\boldsymbol{\mu}_i - \overline{\boldsymbol{\mu}})^t\mathbf{u}\right\|_2$$

$$\leq \sum_{i=1}^k \widehat{w}_i \|(\boldsymbol{\mu}_i - \overline{\boldsymbol{\mu}})\|_2 \max_j |(\boldsymbol{\mu}_j - \overline{\boldsymbol{\mu}}) \cdot \mathbf{u}|$$

$$\leq \sqrt{\sum_{i=1}^k \widehat{w}_i \|(\boldsymbol{\mu}_i - \overline{\boldsymbol{\mu}})\|_2^2} \max_j |(\boldsymbol{\mu}_j - \overline{\boldsymbol{\mu}}) \cdot \mathbf{u}|$$

$$\leq \sqrt{k}\alpha\sigma \max_j |(\boldsymbol{\mu}_j - \overline{\boldsymbol{\mu}}) \cdot \mathbf{u}|.$$

Hence $\sqrt{k}\alpha\sigma \max_i |(\boldsymbol{\mu}_i - \overline{\boldsymbol{\mu}}) \cdot \mathbf{u}| \geq \alpha^2\sigma^2 - 2c(n)$. The lemma follows by substituting the bound on $n$ in $c(n)$. $\square$

We now make a simple observation on Gaussian mixtures.

**Fact 33.** *The samples from a subset of components $A$ of the Gaussian mixture are distributed according to a Gaussian mixture of components $A$ with weights being $w_i' = w_i/(\sum_{j \in A} w_j)$.*

We now prove Lemma 9.

*Proof of Lemma 9.* Observe that we run the recursive clustering at most $n$ times. At every step, the underlying distribution within a cluster is a Gaussian mixture. Let Equations (1), (2) hold with probability $1 - 2\delta$. Let Equations (3) (5) all hold with probability $\geq 1 - \delta'$, where $\delta' = \delta/2n$ at each of $n$ steps. By the union bound the total error is $\leq 2\delta + \delta' \cdot 2n \leq 3\delta$. Since Equations (1), (2) holds, the conditions of Lemmas 7 and 8 hold. Furthermore it can be shown that discarding at most $n\epsilon/4k$ samples at each step does not affect the calculations.

We first show that if $\sqrt{w_i} \|\boldsymbol{\mu}_i - \overline{\boldsymbol{\mu}}(C)\|_2 \geq 25\sqrt{k^3 \log(n^3/\delta)}\sigma$, then the algorithm gets into the loop. Let $w_i'$ be the weight of the component within the cluster and $n' \geq n\epsilon/5k$ be the number of samples in the cluster. Let $\alpha = 25\sqrt{k^3 \log(n^3/\delta)}$. By Fact 33, the components in cluster $C$ have weight $w_i' \geq w_i$. Hence $\sqrt{w_i'} \|\boldsymbol{\mu}_i - \overline{\boldsymbol{\mu}}(C)\|_2 \geq \alpha\sigma$. Since $\sqrt{w_i'} \|\boldsymbol{\mu}_i - \overline{\boldsymbol{\mu}}(C)\|_2 \geq \alpha\sigma$, and by Lemma 8 $\|\boldsymbol{\mu}_i - \overline{\boldsymbol{\mu}}(C)\| \leq 10k\sigma(d \log n^2/\delta)^{1/4}$, we have $w_i' \geq \alpha^2/(100k^2\sqrt{d \log n^2/\delta})$. Hence by lemma 25, $w_i' \geq w_i/2$ and $\sqrt{\widehat{w}_i'} \|\boldsymbol{\mu}_i - \overline{\boldsymbol{\mu}}(C)\|_2 \geq \alpha\sigma/\sqrt{2}$. Hence by Lemma 31 and triangle inequality the largest eigenvalue of $S(C)$ is

$$\geq \alpha^2\sigma^2/2 - c(n') \geq \alpha^2\sigma^2/4 \geq \alpha^2\widehat{\sigma}^2/8 \geq 12\widehat{\sigma}^2 k^3 \log n^2/\delta' = 12\widehat{\sigma}^2 k^3 \log n^3/\delta.$$

Therefore the algorithm gets into the loop.

If $n' \geq n\epsilon/8k^2 \geq c \cdot dk^2 \log \frac{n^3}{\delta}$, then by Lemma 32, there exists a component $i$ such that $|\mathbf{u} \cdot (\boldsymbol{\mu}_i - \overline{\boldsymbol{\mu}}(C))| \geq \sigma(\alpha/\sqrt{2} - 1 - \sqrt{2}/\alpha)/\sqrt{k}$, where $\mathbf{u}$ is the top eigenvector of the first $n\epsilon/4k^2$ samples.

Observe that $\sum_{i \in C} w_i \mathbf{u} \cdot (\boldsymbol{\mu}_i - \overline{\boldsymbol{\mu}}(C)) = 0$ and $\max_i |\mathbf{u} \cdot (\boldsymbol{\mu}_i - \overline{\boldsymbol{\mu}}(C))| \geq \sigma(\alpha/\sqrt{2} - 1 - \sqrt{2}/\alpha)/\sqrt{k}$. Let $\boldsymbol{\mu}_i$ be sorted according to their values of $\mathbf{u} \cdot (\boldsymbol{\mu}_i - \overline{\boldsymbol{\mu}}(C))$, then

$$\max_i |\mathbf{u} \cdot (\boldsymbol{\mu}_i - \boldsymbol{\mu}_{i+1})| \geq \sigma \frac{\alpha/\sqrt{2} - 1 - \sqrt{2}/\alpha}{k^{3/2}} \geq 12\sigma\sqrt{\log \frac{n^3}{\delta}} \geq 9\hat{\sigma}\sqrt{\log \frac{n^3}{\delta}},$$

where the last inequality follows from Lemma 7 and the fact that $d \geq 20 \log n^2/\delta$. For a sample from component $\mathbf{p}_i$, similar to the proof of Lemma 8, by Lemma 15, with probability $\geq 1 - \delta/n^2 k$,

$$\|u \cdot (\mathbf{X}(i) - \boldsymbol{\mu}_i)\| \leq \sigma\sqrt{2\log(n^2 k/\delta)}_2 \leq 2\hat{\sigma}\sqrt{\log(n^2 k/\delta)},$$

where the second inequality follows from Lemma 7. Since there are two components that are far apart by $\geq 9\hat{\sigma}\sqrt{\log \frac{n^2}{\delta}}\hat{\sigma}$ and the maximum distance between a sample and its mean is $\leq 2\hat{\sigma}\sqrt{\log(n^2 k/\delta)}$ and the algorithm divides into at-least two non-empty clusters such that no two samples from the same distribution are clustered into two clusters.

For the second part observe that by the above concentration on $\mathbf{u}$, no two samples from the same component are clustered differently irrespective of the mean separation. Note that we are using the fact that each sample is clustered at most $2k$ times to get the bound on the error probability. The total error probability by the union bound is $\leq 4\delta$. $\qquad\square$

## E.4 Proof of Lemma 10

We show that if the conclusions in Lemmas 9 and 25 holds, then the lemma is satisfied. We also assume that the conclusions in Lemma 31 holds for all the clusters with error probability $\delta' = \delta/k$. By the union bound the total error probability is $\leq 7\delta$.

By Lemma 9 all the components within each cluster satisfy $\sqrt{w_i}\|\boldsymbol{\mu}_i - \overline{\boldsymbol{\mu}}(C)\|_2 \leq 25\sigma\sqrt{k^3 \log(n^3/\delta)}$. Let $n \geq c \cdot dk^9 \epsilon^{-4} \log^2 d/\delta$. For notational convenience let $S(C) = \frac{1}{|C|}\sum_{i=1}^{|C|}(\mathbf{X}(i) - \overline{\boldsymbol{\mu}}(C))(\mathbf{X}(i) - \overline{\boldsymbol{\mu}}(C))^t - \hat{\sigma}^2 \mathbb{I}_d$. Therefore by Lemma 31 for large enough $c$,

$$\left\|S(C) - \frac{n}{|C|}\sum_{i \in C}\hat{w}_i(\boldsymbol{\mu}_i - \overline{\boldsymbol{\mu}}(C))(\boldsymbol{\mu}_i - \overline{\boldsymbol{\mu}}(C))^t\right\| \leq \frac{\epsilon^2 \sigma^2}{1000 k^2}\frac{n}{|C|}.$$

Let $\mathbf{v}_1, \mathbf{v}_2, \ldots \mathbf{v}_{k-1}$ be the top eigenvectors of $\frac{1}{|C|}\sum_{i \in C} w_i(\boldsymbol{\mu}_i - \overline{\boldsymbol{\mu}}(C))(\boldsymbol{\mu}_i - \overline{\boldsymbol{\mu}}(C))^t$. Let $\eta_i = \sqrt{\hat{w}_i'}\|\boldsymbol{\mu}_i - \overline{\boldsymbol{\mu}}(C)\|_2 = \sqrt{\hat{w}_i}\sqrt{\frac{n}{|C|}}\|\boldsymbol{\mu}_i - \overline{\boldsymbol{\mu}}(C)\|_2$. Let $\boldsymbol{\Delta}_i = \frac{\boldsymbol{\mu}_i - \overline{\boldsymbol{\mu}}(C)}{\|(\boldsymbol{\mu}_i - \overline{\boldsymbol{\mu}}(C))\|_2}$. Therefore,

$$\sum_{i \in C}\frac{n}{|C|}\sum_{i \in C}\hat{w}_i(\boldsymbol{\mu}_i - \overline{\boldsymbol{\mu}}(C))(\boldsymbol{\mu}_i - \overline{\boldsymbol{\mu}}(C))^t = \sum_{i \in C}\eta_i^2 \boldsymbol{\Delta}_i \boldsymbol{\Delta}_i^t.$$

Hence by Lemma 21, the projection of $\boldsymbol{\Delta}_i$ on the space orthogonal to top $k-1$ eigenvectors of $S(C)$ is

$$\leq \sqrt{\frac{\epsilon^2 \sigma^2}{1000 k^2}\frac{n}{|C|}\frac{1}{\eta_i}} \leq \frac{\epsilon\sigma}{16\sqrt{\hat{w}_i}\|\boldsymbol{\mu}_i - \overline{\boldsymbol{\mu}}(C)\|_2 k} \leq \frac{\epsilon\sigma}{8\sqrt{2}\sqrt{w_i}\|\boldsymbol{\mu}_i - \overline{\boldsymbol{\mu}}(C)\|_2 k}.$$

The last inequality follows from the bound on $\hat{w}_i$ in Lemma 25.

## E.5 Proof of Theorem 11

We show that the theorem holds if the conclusions in Lemmas 10 and 27 holds with error probability $\delta' = \delta/k$. Since in the proof of Lemma 10, the probability that Lemma 9 holds is included, Lemma 9 also holds with the same probability. Since there are at most $k$ clusters, by the union bound the total error probability is $\leq 9\delta$.

For every component $i$, we show that there is a choice of mean vector and weight in the search step such that $w_i D(\mathbf{p}_i, \hat{\mathbf{p}}_i) \leq \epsilon/2k$ and $|w_i - \hat{w}_i| \leq \epsilon/4k$. That would imply that there is a $\hat{\mathbf{f}}$ during the search such that

$$D(\mathbf{f}, \hat{\mathbf{f}}) \leq \sum_C \sum_{i \in C} w_i D(\mathbf{p}_i, \hat{\mathbf{p}}_i) + 2\sum_{i=1}^{k-1}|w_i - \hat{w}_i| \leq \frac{\epsilon}{2k} + \frac{\epsilon}{2k} = \epsilon.$$

Since the weights are gridded by $\epsilon/4k$, there exists a $\hat{w}_i$ such that $|w_i - \hat{w}_i| \leq \epsilon/4k$. We now show that there exists a choice of mean vector such that $w_i D(\mathbf{p}_i, \hat{\mathbf{p}}_i) \leq \epsilon/2k$. Note that if a component has weight $\leq \epsilon/4k$, the above inequality follows immediately. Therefore we only look at those components with $w_i \geq \epsilon/4k$, by Lemma 25, for such components $\hat{w}_i \geq \epsilon/5k$ and therefore we only look at clusters such that $|C| \geq n\epsilon/5k$. By Lemmas 14 and for any $i$,

$$D(\mathbf{p}_i, \hat{\mathbf{p}}_i)^2 \leq 2\sum_{j=1}^{d} \frac{(\mu_{i,j} - \hat{\mu}_{i,j})^2}{\sigma^2} + 8d\frac{(\sigma^2 - \hat{\sigma}^2)^2}{\sigma^4}.$$

Note that since we are discarding at most $n\epsilon/8k^2$ random samples at each step. A total number of $\leq n\epsilon/8k$ random samples are discarded. It can be shown that this does not affect our calculations and we ignore it in this proof. By Lemma 7, the first estimate of $\sigma^2$ satisfies $|\hat{\sigma}^2 - \sigma^2| \leq 2.5\sigma^2\sqrt{\frac{\log n^2/\delta}{d}}$. Hence while searching over values of $\hat{\sigma}^2$, there exist one such that $|\sigma'^2 - \sigma^2| \leq \epsilon\sigma^2/\sqrt{64dk^2}$. Hence,

$$D(\mathbf{p}_i, \hat{\mathbf{p}}_i)^2 \leq 2\frac{\|\boldsymbol{\mu}_i - \hat{\boldsymbol{\mu}}_i\|_2^2}{\sigma^2} + \frac{\epsilon^2}{8k^2}.$$

Therefore if we show that there is a mean vector $\hat{\boldsymbol{\mu}}_i$ during the search such that $\|\boldsymbol{\mu}_i - \hat{\boldsymbol{\mu}}_i\|_2 \leq \epsilon\sigma/\sqrt{16k^2\hat{w}_i}$, that would prove the Lemma. By triangle inequality,

$$\|\boldsymbol{\mu}_i - \hat{\boldsymbol{\mu}}_i\|_2 \leq \left\|\overline{\boldsymbol{\mu}}(C) - \hat{\overline{\boldsymbol{\mu}}}(C)\right\|_2 + \left\|\boldsymbol{\mu}_i - \overline{\boldsymbol{\mu}}(C) - (\hat{\boldsymbol{\mu}}_i - \hat{\overline{\boldsymbol{\mu}}}(C))\right\|_2.$$

By Lemma 27 for large enough $n$,

$$\left\|\overline{\boldsymbol{\mu}}(C) - \hat{\overline{\boldsymbol{\mu}}}(C)\right\|_2 \leq c\sigma\sqrt{\frac{dk\log^2 n^2/\delta}{|C|}} \leq \frac{\epsilon\sigma}{8k\sqrt{w_i}}.$$

The second inequality follows from the bound on $n$ and the fact that $|C| \geq n\hat{w}_i$. Since $w_i \geq \epsilon/4k$, by Lemma 25, $\hat{w}_i \geq w_i/2$, we have

$$\|\boldsymbol{\mu}_i - \hat{\boldsymbol{\mu}}_i\|_2 \leq \left\|\boldsymbol{\mu}_i - \overline{\boldsymbol{\mu}}(C) - (\hat{\boldsymbol{\mu}}_i - \hat{\overline{\boldsymbol{\mu}}}(C))\right\|_2 + \frac{\epsilon\sigma}{8k\sqrt{w_i}}.$$

Let $\mathbf{u}_1 \ldots \mathbf{u}_{k-1}$ are the top eigenvectors the sample covariance matrix of cluster $C$. We now prove that during the search, there is a vector of the form $\sum_{j=1}^{k-1} g_j\epsilon_g\hat{\sigma}\mathbf{u}_j$ such that $\left\|\boldsymbol{\mu}_i - \overline{\boldsymbol{\mu}}(C) - \sum_{j=1}^{k-1} g_j\epsilon_g\hat{\sigma}\mathbf{u}_j\right\|_2 \leq \frac{\epsilon\sigma}{8k\sqrt{w_i}}$, during the search, thus proving the lemma. Let $\eta_i = \sqrt{w_i}\|\boldsymbol{\mu}_i - \overline{\boldsymbol{\mu}}(C)\|_2$. By Lemma 10, there are set of coefficients $\alpha_i$ such that

$$\frac{\boldsymbol{\mu}_i - \overline{\boldsymbol{\mu}}(C)}{\|\boldsymbol{\mu}_i - \overline{\boldsymbol{\mu}}(C)\|_2} = \sum_{j=1}^{k-1} \alpha_j\mathbf{u}_j + \sqrt{1 - \|\alpha\|^2}\mathbf{u}',$$

where $\mathbf{u}'$ is perpendicular to $\mathbf{u}_1 \ldots \mathbf{u}_{k-1}$ and $\sqrt{1 - \|\alpha\|^2} \leq \epsilon\sigma/(8\sqrt{2}\eta_i k)$. Hence, we have

$$\boldsymbol{\mu}_i - \overline{\boldsymbol{\mu}}(C) = \sum_{j=1}^{k-1} \|\boldsymbol{\mu}_i - \overline{\boldsymbol{\mu}}(C)\|_2 \alpha_j\mathbf{u}_j + \|\boldsymbol{\mu}_i - \overline{\boldsymbol{\mu}}(C)\|_2 \sqrt{1 - \|\alpha\|_2^2}\mathbf{u}',$$

Since $w_i \geq \epsilon/4k$ and by Lemma 9, $\eta_i \leq 25\sqrt{k^3}\sigma\log(n^3/\delta)$, and $\|\boldsymbol{\mu}_i - \overline{\boldsymbol{\mu}}(C)\|_2 \leq 100\sqrt{k^4\epsilon^{-1}}\sigma\log(n^3/\delta)$. Therefore $\exists g_j$ such that $|g_j\hat{\sigma} - \alpha_j| \leq \epsilon_g\hat{\sigma}$ on each eigenvector. Hence,

$$w_i\left\|\boldsymbol{\mu}_i - \overline{\boldsymbol{\mu}}(C) - \sum_{i=1}^{k-1} g_j\epsilon_g\hat{\sigma}\mathbf{u}_j\right\|_2^2 \leq w_i k\epsilon_g^2\hat{\sigma}^2 + w_i\|\boldsymbol{\mu}_i - \overline{\boldsymbol{\mu}}(C)\|_2^2 (1 - \|\alpha\|^2)$$

$$\leq k\epsilon_g^2\hat{\sigma}^2 + \eta_i^2 \frac{\epsilon^2\sigma^2}{128\eta_i^2 k^2}$$

$$\leq \frac{\epsilon^2\sigma^2}{128k^2} + \frac{\epsilon^2\sigma^2}{128k^2} \leq \frac{\epsilon^2\sigma^2}{64k^2}.$$

The last inequality follows by Lemma 7 and the fact that $\epsilon_g \leq \epsilon/16k^{3/2}$, and hence the theorem. The run time can be easily computed by retracing the steps of the algorithm and using an efficient implementation of single-linkage.

# F Mixtures with unequal variances

In this section, we outline the analysis for the case when the components have different variances.

The main difference would be the coarse clustering algorithm which we describe now. The algorithm repeatedly finds components with smallest variances and clusters samples such that within each cluster the variances differ by a factor of $1 + \widetilde{\mathcal{O}}(1/\sqrt{d})$ and the means are close-by. However, two subtleties arise.

**Randomized thresholding:** Suppose we fix a threshold for clustering in step 3 of the coarse clustering algorithm, then there might be a component whose average distance from $\mathbf{x}(a)$ or $\mathbf{x}(b)$ is exactly the threshold and due to randomness in samples, few samples can lie in one cluster and few can lie on the other. We overcome this, by choosing a random threshold, thus making it unlikely that there is a component with average distance at the threshold.

**Components with single sample:** If two samples are from the same component $i$, then their squared-distance concentrates around $2d\sigma_i^2$. We can use this fact to estimate the variance. However if there is only one sample from a component, we cannot estimate its variance and moreover it can affect the calculations of other components. Hence in Step 4, we find such components and discard the corresponding samples.

---

**Generalized coarse clustering:** Let $\alpha = 4\sqrt{\log(n^2/\delta)/d}$. Initialize $C$ to the set of all samples. Repeat the following $k$ times.

1. Find threshold $\mathtt{t} = \min_{a \neq b, a, b \in C} \|\mathbf{x}(a) - \mathbf{x}(b)\|_2$. Let $a$ and $b$ be the indices that achieve this minimum.

2. Let r be a uniform random variable between $10$ and $4000k^2$.

3. Find the set of samples $C_1$ that are at a distance $\leq \mathtt{t}\sqrt{(1 + \alpha \mathtt{r})}$ from either $\mathbf{x}(a)$ or $\mathbf{x}(b)$.

4. If the $\max_{c, d \in C_1} \|\mathbf{x}(c) - \mathbf{x}(d)\|_2^2 > \mathtt{t}\sqrt{(1 + 50\alpha \mathtt{r})}$, discard $\mathbf{x}(a)$, $\mathbf{x}(b)$ and the samples that achieve the maximum, else declare $C_1$ as a new cluster and remove samples in $C_1$ from $C$.

---

The rest of the analysis is similar to the case with equal variances. We now outline analysis for **Generalized coarse clustering**. We first show an auxiliary concentration inequality that helps us prove the rest of the results.

**Lemma 34.** *Given $n$ samples from a set of Gaussian distributions, with probability $\geq 1 - 2\delta$, for every pair of samples $\mathbf{X} \sim N(\boldsymbol{\mu}_1, \sigma_1^2 \mathbb{I}_d)$ and $\mathbf{Y} \sim N(\boldsymbol{\mu}_2, \sigma_2^2 \mathbb{I}_d)$,*

$$1 - 4\sqrt{\frac{\log \frac{n^2}{\delta}}{d}} \leq \frac{\|\mathbf{X} - \mathbf{Y}\|_2^2}{d(\sigma_1^2 + \sigma_2^2) + \|\boldsymbol{\mu}_1 - \boldsymbol{\mu}_2\|_2^2} \leq 1 + 4\sqrt{\frac{\log \frac{n^2}{\delta}}{d}}. \tag{7}$$

*Proof.* Since $\mathbf{X}$ and $\mathbf{Y}$ are Gaussians, $\mathbf{X} - \mathbf{Y}$ is distributed $N(\boldsymbol{\mu}_1 - \boldsymbol{\mu}_2, (\sigma_1^2 + \sigma_2^2)\mathbb{I}_d)$. Therefore substituting $t = \log \frac{n^2}{\delta}$ in Lemma 17, with probability $1 - 4\delta/n^2$,

$$\|\mathbf{X} - \mathbf{Y}\|_2^2 \geq d(\sigma_1^2 + \sigma_2^2) - 2(\sigma_1^2 + \sigma_2^2)\sqrt{d \log \frac{n^2}{\delta}} + \|\boldsymbol{\mu}_1 - \boldsymbol{\mu}_2\|_2^2 - 2\sqrt{\sigma_1^2 + \sigma_2^2}\|\boldsymbol{\mu}_1 - \boldsymbol{\mu}_2\|_2 \sqrt{\log \frac{n^2}{\delta}}.$$

and

$$\|\mathbf{X} - \mathbf{Y}\|_2^2 \leq d(\sigma_1^2 + \sigma_2^2) + 2(\sigma_1^2 + \sigma_2^2)\sqrt{d \log \frac{n^2}{\delta}} + \|\boldsymbol{\mu}_1 - \boldsymbol{\mu}_2\|_2^2$$
$$+ 2\sqrt{\sigma_1^2 + \sigma_2^2}\|\boldsymbol{\mu}_1 - \boldsymbol{\mu}_2\|_2 \sqrt{\log \frac{n^2}{\delta}} + 2(\sigma_1^2 + \sigma_2^2)\log \frac{n^2}{\delta}.$$

There are $\binom{n}{2}$ pairs and the error probability follows by the union bound. Dividing the bounds by $d(\sigma_1^2 + \sigma_2^2) + \|\boldsymbol{\mu}_1 - \boldsymbol{\mu}_2\|_2^2$ and using the arithmetic-geometric mean inequality we get

$$1 - 3\sqrt{\frac{\log \frac{n^2}{\delta}}{d}} \leq \frac{\|\mathbf{X} - \mathbf{Y}\|_2^2}{d(\sigma_1^2 + \sigma_2^2) + \|\boldsymbol{\mu}_1 - \boldsymbol{\mu}_2\|_2^2} \leq 1 + 3\sqrt{\frac{\log \frac{n^2}{\delta}}{d}} + 2\frac{\log \frac{n^2}{\delta}}{d}.$$

Using $d \geq 20 \log \frac{n^2}{\delta}$ proves the lemma. $\qquad \square$

We now show a few properties of Coarse clustering. In particular, we show that

- There is no mis-clustering.
- After $k$ steps of iteration, all the samples would be clustered.
- The means and variances of all components within any cluster are close to each other.

Let $\alpha \stackrel{\text{def}}{=} 4\sqrt{\frac{\log \frac{n^2}{\delta}}{d}}$. For the rest of the proof we assume that $d \geq 4000 \log(n^2/\delta)$, thus $\alpha \leq 1/10$. We first show that the probability of mis-clustering is $\leq 1/100$.

**Lemma 35.** *If Equation* (7) *holds, then after coarse clustering algorithm, with probability $\geq 99/100$, all the samples from each component will be in the same cluster.*

*Proof.* Without loss of generality, let $\mathbf{x}(a)$ be from component $1$ and $\mathbf{x}(b)$ be from component $2$. If for all components $i$ and $j \in \{1,2\}$ if $\left(d(\sigma_j^2 + \sigma_i^2) + \|\boldsymbol{\mu}_j - \boldsymbol{\mu}_i\|_2^2\right)(1 + \alpha) < \mathsf{t}^2(1 + \alpha \mathsf{r})$ or $\left(d(\sigma_j^2 + \sigma_i^2) + \|\boldsymbol{\mu}_j - \boldsymbol{\mu}_i\|_2^2\right)(1 - \alpha) > \mathsf{t}^2(1 + \alpha \mathsf{r})$, then by Equation (7) the pairwise distances concentrate and all the samples would be clustered without any error. Hence the error probability is

$$\Pr\left(\exists i,j \text{ s.t. } \mathsf{t}^2(1 + \alpha\mathsf{r}) \in \left[\left(d(\sigma_j^2 + \sigma_i^2) + \|\boldsymbol{\mu}_j - \boldsymbol{\mu}_i\|_2^2\right)(1 - \alpha), \left(d(\sigma_j^2 + \sigma_i^2) + \|\boldsymbol{\mu}_j - \boldsymbol{\mu}_i\|_2^2\right)(1 + \alpha)\right]\right).$$

For a given $i, j$, this probability is

$$\leq \frac{2}{4000\mathsf{t}^2 k^2 - 10}\left(d(\sigma_j^2 + \sigma_i^2) + \|\boldsymbol{\mu}_j - \boldsymbol{\mu}_i\|_2^2\right)\mathbb{1}\left(\left(d(\sigma_j^2 + \sigma_i^2) + \|\boldsymbol{\mu}_j - \boldsymbol{\mu}_i\|_2^2\right)(1 - \alpha) \leq \mathsf{t}^2(1 + 4000k^2\alpha)\right).$$

Since $d \geq c \cdot k^4 \log \frac{n^2}{\delta}$ for a large enough constant $c$, we have $1 + 4000k^2\alpha \leq 2$. Hence, the above probability is $\leq \frac{4}{3990(1-\alpha)k^2} \leq \frac{1}{997(1-\alpha)k^2}$. Since $\alpha \leq 1/10$, this is $\leq \frac{1}{200k^2}$. By the union bound over all possible components $i$, $j$, the error probability is $\leq \frac{1}{100k}$. Since we run the algorithm $k$ times, by the union bound the total error probability is $\leq \frac{1}{100}$. $\qquad \square$

**Lemma 36.** *If Equation* (7) *holds and there is no mis-clustering, and a cluster is created at any of the $k$ steps , then for each pair of components $i, j$ in that cluster with $\hat{w}_i, \hat{w}_j \geq 2/n$, $2d\sigma_i^2 \in \left[\mathsf{t}^2(1 - \alpha), \mathsf{t}^2(1 + 56\alpha\mathsf{r})\right]$ and $\|\boldsymbol{\mu}_i - \boldsymbol{\mu}_j\|_2^2 \leq c \cdot k^2 \mathsf{t}^2 \alpha$ for some constant $c$. Furthermore, for every other component $l$, $\|\boldsymbol{\mu}_i - \boldsymbol{\mu}_l\|_2^2 + \sigma_i^2 \leq c \cdot \mathsf{t}^2$.*

*Proof.* The square of the maximum separation between any two samples in a cluster is $\leq \mathsf{t}^2(1 + 50\alpha\mathsf{r})$ and the points are clustered correctly. Let $i$ be a component such that $\hat{w}_i \geq 2/n$. Let $\mathbf{x}(g)$ and $\mathbf{x}(h)$ be two samples from component $i$, then

$$2d\sigma_i^2(1 - \alpha) \leq \|\mathbf{x}(g) - \mathbf{x}(h)\|_2^2 \leq \mathsf{t}^2(1 + 50\alpha\mathsf{r}),$$

where the first inequality follows from Equation (7). Hence, $2d\sigma_i^2 \leq \mathsf{t}^2(1 + 50\alpha\mathsf{r})/(1 - \alpha) \leq \mathsf{t}^2(1 + 56\alpha\mathsf{r})$. Furthermore, since $\mathbf{x}(g)$ and $\mathbf{x}(h)$ has pairwise distance $\geq \mathsf{t}$, by Equation (7),

$$2d\sigma_i^2(1 + \alpha) \geq \|\mathbf{x}(g) - \mathbf{x}(h)\|_2^2 \geq \mathsf{t}^2,$$

and hence $2d\sigma_i^2 \geq \mathsf{t}^2(1 - \alpha)$.

For two samples $\mathbf{x}(g)$ and $\mathbf{x}(h)$ generated by components $i$ and $j$, we have,

$$\mathsf{t}^2(1 + 50\alpha\mathsf{r}) \overset{(a)}{\geq} \|\mathbf{x}(g) - \mathbf{x}(h)\|_2^2$$
$$\overset{(b)}{\geq} \left(d(\sigma_i^2 + \sigma_j^2) + \|\boldsymbol{\mu}_i - \boldsymbol{\mu}_j\|_2^2\right)(1 - \alpha)$$
$$\overset{(c)}{\geq} \mathsf{t}^2(1 - 2\alpha) + \|\boldsymbol{\mu}_i - \boldsymbol{\mu}_j\|_2^2(1 - \alpha),$$

where $(a)$ follows from the fact that the maximum separation between two samples is $\leq \mathsf{t}^2(1 + 50\alpha\mathsf{r})$, Equation (7) implies $(b)$, and $(c)$ follows from first part of the lemma. Hence, we have $\left\|\boldsymbol{\mu}_i - \boldsymbol{\mu}_j\right\|_2^2 \leq \mathsf{t}^2(50\alpha\mathsf{r} + 2\alpha)/(1 - \alpha) \leq \mathsf{t}^2(3 \cdot 10^6 \alpha k^2)$.

Let $\mathbf{x}(g)$ and $\mathbf{x}(h)$ be from components $i$ and $l$ respectively. Similar to the first two parts of the lemma we have, maximum separation between any two samples is

$$\mathsf{t}^2(1+50\alpha\mathsf{r}) \geq \|\mathbf{x}(g) - \mathbf{x}(h)\|_2^2 \geq \left(d(\sigma_i^2 + \sigma_l^2) + \|\boldsymbol{\mu}_i - \boldsymbol{\mu}_l\|_2^2\right)(1-\alpha) \geq \left(d\sigma_l^2 + \|\boldsymbol{\mu}_i - \boldsymbol{\mu}_l\|_2^2\right)(1-\alpha).$$

Hence $d\sigma_l^2 + \|\boldsymbol{\mu}_i - \boldsymbol{\mu}_l\|_2^2 \leq \mathsf{t}^2(1 + 50\alpha\mathsf{r})/(1 - \alpha) \leq c \cdot \mathsf{t}^2$, for some constant $c$. The last part follows from the assumption that $d = \Omega(k^4 \log \frac{n^2}{\delta})$. $\qquad\square$

**Lemma 37.** *If Equation* (7) *holds and there is no mis-clustering, at end of the generalized coarse clustering* $|C| = 0$.

*Proof.* We show that if $C$ is non-empty, at each iteration the number of components in $C$ decreases by at least one. Since there is no mis-clustering, if we create a cluster at a particular iteration, it would contain all the samples from at least one component and hence the number of components in $C$ reduces by one. We now show that if we discard four samples, at least one of them would be a unique sample from its component ($\hat{w}_i = 1/n$) and hence discarding it would reduce the number of components by one.

Let $\mathbf{x}(a), \mathbf{x}(b)$ be the two samples that attain the minimum and without loss of generality let the corresponding components be 1 and 2. Let $\mathbf{x}(c), \mathbf{x}(d)$ be the two samples that achieve the maximum and $i, j$ be their corresponding components. We now show that if $\min(\hat{w}_1, \hat{w}_2, \hat{w}_i, \hat{w}_j) \geq 2/n$, then the samples would not be discarded thus proving our claim. By Equation (7),

$$d(\sigma_1^2 + \sigma_2^2) + \|\boldsymbol{\mu}_1 - \boldsymbol{\mu}_2\|_2^2 \leq \frac{\|\mathbf{x}(a) - \mathbf{x}(b)\|_2^2}{1 - \alpha} \leq \mathsf{t}^2(1 + 3\alpha),$$

and since two samples from component 1 or 2 did not achieve the minimum, $2d\sigma_2^2 \geq \mathsf{t}^2(1 - \alpha)$ and $2d\sigma_1^2 \geq \mathsf{t}^2(1 - \alpha)$. Rearranging and substituting in the three equations we get, $\|\boldsymbol{\mu}_1 - \boldsymbol{\mu}_2\|_2^2 \leq 4\mathsf{t}^2\alpha$, $2d\sigma_1^2 \leq \mathsf{t}^2(1 + 7\alpha)$ and $2d\sigma_2^2 \leq \mathsf{t}^2(1 + 7\alpha)$. Without loss of generality, let $\mathbf{x}(c)$ be included in $C_1$ because $\mathbf{x}(c)$ was close to $\mathbf{x}(a)$.

$$d(\sigma_1^2 + \sigma_i^2) + \|\boldsymbol{\mu}_1 - \boldsymbol{\mu}_i\|_2^2 \leq \frac{\|\mathbf{x}(a) - \mathbf{x}(c)\|_2^2}{1 - \alpha} \leq \mathsf{t}^2(1 + 3\alpha\mathsf{r}),$$

and furthermore two samples from components $i$ or 1 did not achieve minimum and hence, $2d\sigma_i^2 \geq \mathsf{t}^2(1 - \alpha)$ and $2d\sigma_1^2 \geq \mathsf{t}^2(1 - \alpha)$. Solving, we get $2d\sigma_i^2 \leq \mathsf{t}^2(1 + 7\alpha\mathsf{r})$ and $\|\boldsymbol{\mu}_1 - \boldsymbol{\mu}_i\|_2^2 \leq 4\mathsf{t}^2\alpha\mathsf{r}$. Similarly, $2d\sigma_j^2 \leq \mathsf{t}^2(1 + 7\alpha\mathsf{r})$ and $\left\|\boldsymbol{\mu}_l - \boldsymbol{\mu}_j\right\|_2^2 \leq 4\mathsf{t}^2\alpha\mathsf{r}$, for some $l \in \{1, 2\}$. We now have all the inequalities necessary to show that $\|\mathbf{x}(c) - \mathbf{x}(d)\|_2^2 \leq \mathsf{t}^2(1 + 50\alpha\mathsf{r})$ and hence would not be discarded.

$$\begin{aligned}
\|\mathbf{x}(c) - \mathbf{x}(d)\|_2^2 &\overset{(a)}{\leq} \left(d(\sigma_i^2 + \sigma_j^2) + \left\|\boldsymbol{\mu}_i - \boldsymbol{\mu}_j\right\|_2^2\right)(1 + \alpha) \\
&\overset{(b)}{\leq} \mathsf{t}^2(1 + 7\alpha\mathsf{r}) + \left\|\boldsymbol{\mu}_i - \boldsymbol{\mu}_1 + \boldsymbol{\mu}_1 - \boldsymbol{\mu}_l + \boldsymbol{\mu}_l - \boldsymbol{\mu}_j\right\|_2^2 \\
&\overset{(c)}{\leq} \mathsf{t}^2(1 + 7\alpha\mathsf{r}) + 3(\|\boldsymbol{\mu}_i - \boldsymbol{\mu}_1\|_2^2 + \|\boldsymbol{\mu}_1 - \boldsymbol{\mu}_l\|_2^2 + \left\|\boldsymbol{\mu}_l - \boldsymbol{\mu}_j\right\|_2^2) \\
&\overset{(d)}{\leq} \mathsf{t}^2(1 + 50\alpha\mathsf{r}).
\end{aligned}$$

$(a)$ follows from Equation (7). $(b)$ follows from the bounds on $\sigma_i^2$ and $\sigma_j^2$. $(c)$ follows from Cauchy-Schwarz inequality and $(d)$ the bounds on the difference of means which we have shown implies $(d)$. $\qquad\square$

The above four lemmas immediately yields,

**Lemma 38.** *After coarse clustering, the algorithm divides the samples into clusters such that with probability* $\geq 99/100 - 2\delta$,

- *There is no mis-clustering.*

- *For any pair of components $i, j$ within a cluster with $\hat{w}_i, \hat{w}_j \geq 2/n$, the variances lie within a factor of $1 \pm 56\alpha r$ around $t^2$ and $\left\|\boldsymbol{\mu}_i - \boldsymbol{\mu}_j\right\|_2^2 \leq \mathcal{O}(t^2\alpha^2)$.*

- *For every component $i$ within a cluster $C$, $\|\boldsymbol{\mu}_i - \overline{\boldsymbol{\mu}}(C)\|_2^2 + d\sigma_i^2 \leq \mathcal{O}(t^2)$.*

It can be shown that once the conclusions in Lemma 38 holds, then the performance of recursive clustering algorithm would be same as Lemma 9 upto constants. The only modification is the computation of $\hat{\sigma}^2(C)$ which is given by

$$\hat{\sigma}^2(C) = \frac{1}{|C|(|C|-1)} \sum_{a,b \in C} \frac{1}{2d} \|\mathbf{x}(a) - \mathbf{x}(b)\|_2^2.$$

By Lemma 38, out of $\binom{|C|}{2}$ pairs at most $k|C|$ would have distances away from $t^2$. It can be shown that this does not affect the analysis.

Finally for the exhaustive search, instead of just substituting a single $\sigma'$, we try out all possible combinations of $\sigma'(C)$ for each cluster $C$, where $\sigma'(C) \in \hat{\sigma}^2(1 + i\epsilon/d\sqrt{128dk^2})$, $\forall -L' < i \leq L'\}$, where $L' = \frac{32k\sqrt{\log n^2/\delta}}{\epsilon}$. Note that since we are searching over $k$ different variances instead of just one, the number of candidate mixtures increases by and hence the time complexity. The time complexity for unequal variances can be shown to be

$$\mathcal{O}\left(n^2 d \log n + d\left(\frac{k^7}{\epsilon^3} \log^2 \frac{d}{\delta}\right)^{\frac{k^2}{2}} \left(\frac{k\sqrt{\log d/\delta}}{\epsilon}\right)^k\right).$$

Note that even though our error probability is $1/100 + 2\delta$, and is not arbitrarily close to $0$, we can repeat the entire algorithm $\mathcal{O}(\log \frac{1}{\delta'})$ times and run SCHEFFE on the resulting components to find the closest one. By the Chernoff bound, the error probability of this new estimator would be $\leq \delta'$.