[Reviews · NeurIPS 2014]

Submitted by Assigned_Reviewer_9

This paper considers PAC learning of mixture spherical Gaussian distributions. Via exploiting the spherical structure of the data (in particular, the assumption that all covariance matrices of all components are diagonal), the authors are able to derive efficient algorithms to estimate the means and the standard deviations. The mixtures can then be derived using a modified Scheffe's method,

This paper is, in general, very well written. The intuition, the proposed method, and the theoretical justifications are all stated clearly and reader-friendly. I can follow the main contents very easily. The related literature seems to be well cited (I am not an expert in this area, so not very confirmative). The results look novel, correct, and important. Overall, I enjoy reading it a lot.

However, there are still some issues need to be addressed:
(1) It looks like that the whole paper is focused on the setting that k, the number of components, is known. It is a little bit surprising that the authors did not discuss the situation when k is unknown. Is it trivial to estimate it? At least some heuristics are preferred.
(2) By assuming sphericity on the components, The results look like quite restrictive in real applications. Some discussions on retionality of sphericity assumption, or a generalization to more general cases, are desired.
(3) Technically, the bound on delta, provided in line 288, is a little bit weird. For example, if d is fixed, this delta can be as large as 1/3.

Minor comments:
(1) Line 47-48, rephrase the sentence.
(2) line 247, S looks more like a covariance-type matrix.
(3) line 270, "necessary" -> "necessarily"?

Summary: This paper discusses efficiently estimating the Gaussian mixtures. The results are new and important. I enjoy reading this paper.

Submitted by Assigned_Reviewer_11

Overall, this seems like a reasonable paper, though the approach does
not contain any especially provoking or new ideas.

My main concern is that this paper seems to be basically turning the
crank; these results were known or suspected by most people (perhaps
with the linear in d replaced by d^2), and the various algorithmic
components (considering the subspace spanned by the mean vectors,
using some clustering to partition far-apart components, using the
closest pair of points as a proxy for the scale of the variance,
running scheffe-like algorithms/tournaments over a gridding of the
space) are all approaches that have been used for this problem for a
while. In my view, the real frontier on this problem seems to be the
algorithmic (as opposed to information theoretic) question, perhaps in
the agnostic setting in which practically efficient algorithms might
be possible via very new approaches. Pinning down tighter results in
the spherical gaussians case without introducing new techniques (that
might apply more generally) seems to be a little less exciting.
Summary: The main results of this paper are establishing bounds on the sample
complexity of learning spherical GMMs. [The runtimes of the
algorithms, while maybe slightly better than previous algorithms, are
all exponential in the number of components, and proceed by
brute-force enumeration over an appropriately gridded space....]

Perhaps the nicest idea in the paper is to consider the top k
eigenvectors of the sample covariance, to use as a proxy for the
k-dimensional subspace spanned by the component means. This improves
upon previous approaches that looked at the span of columns of the
covariance (which requires that the whole covariance be
well-approximated, rather than just the top k-subspace). This allows
for a linear in the dimension sample dependence, and poly(k)
dependence.

The actual algorithm proceeds by some clustering phase, to guarantee
that the above concentration will not be ruined by some
far-off-components, and then a gridding of the space of potential
distributions. Finally, the scheffe algorithm is used, modified so as
to run in a near-linear as opposed to quadratic time to pick the best
of the gridded space of possible distributions. This establishes a
sample complexity logarithmic in the size of the set of potential
hypotheses.

One thing worth considering:
I am not sure the chronology of this work, versus [11] (Daskalakis,
Kamath). The following sentence starting line 118 is a little
strange:
"We note that independently and concurrently with this work [11]
showed that mixtures of two one-dimensional Gaussians can be learnt
with O~(eps^−2) samples and in time O~(eps^-5). Combining with some of
the techniques in this paper, they extend their algorithm to mixtures
of k Gaussians, and reduce the exponent to 3k − 1."

Submitted by Assigned_Reviewer_23

This paper egregiously violated the NIPS formatting rules: the margins were shrank considerably. The overall paper far exceeds the 8 pages limit if the margins were restored to their correct size.

The overall paper and its results seem solid. The ideas are clearly described. I did not read the proofs but the intuition provided by the paper makes sense.

Summary: The paper seems OK but I did not review it carefully because of serious format violations.

Submitted by Assigned_Reviewer_24

The main result of this paper is an algorithm for properly PAC-learning a mixture of Gaussians in high dimensions. More precisely, the authors give an algorithm that takes a small number of samples from a mixture of k spherical Gaussians in d dimensions (k is assumed to be known) and outputs another mixture of k Gaussians that’s within \epsilon total variation distance. The merit of the algorithm is in the small number of samples it uses: linear in d and small polynomial dependence on k and 1/\epsilon. The time complexity is exponential in k.

For Gaussian mixtures, one normally wants to learn the parameters of the component distributions of the mixture. The paper under review addresses the weaker requirement of properly PAC-learning the given mixture. Why is this interesting? The authors do not try to answer this. But one could try to imagine situations where one needs to generate more samples from the given distribution and there PAC-learning mixtures could be useful. But then even improper learning might suffice for that purpose. So, as far as I can see, the motivation for studying the problem is somewhat weak, but given the importance of mixtures of Gaussians I think it’s interesting.

The techniques build upon and combine previous work (Scheffe’s algorithm, spectral clustering, PCA) in an interesting way. I have not verified the proofs (all in the appendices) but I find the theorems believable.
Summary: The paper has interesting ideas and will likely stimulate further work.
Author Feedback
Author rebuttal: We thank the reviewers for their time and comments. We address the comments of each of the reviewers below.

------------

Firstly, we would like thank Assigned_Reviewer_23 for pointing out that we have violated the margin conditions. We completely agree with the problem with margins. It was a big yet honest mistake, and we sincerely apologize for it. We had a version in our internal drafts and when we incorporated the NIPS style, we forgot to remove a usepackage{fullpage} which caused the mistake. We would rectify the mistake in the final camera ready version, if accepted, and hope that this does not affect the decision of the paper.

------------

Assigned_Reviewer_24 raises the question of motivation for PAC learning. We study PAC learning as it is the simplest and most natural statistical question. Furthermore, the broader problem of parameter learning can be treated as a combination of PAC learning (learning a distribution) and identifiability: if two distributions are similar in l1 distance - then so are their parameters. In this paper we address the first issue of PAC learning.

As the reviewer suggests, PAC learning can be used to generate further samples from the underlying distribution, but for this application improper learning suffices. He therefore asks the necessity of proper PAC learning. We are not aware of efficient improper learning to efficiently high dimensional mixtures.
Recently, in STOC 2014, ``Efficient Density Estimation via Piecewise Polynomial Approximation'' provides an efficient improper learning algorithm for one dimensional case and it is not clear if those results can be extended to higher dimensions.

-------------
Assigned_Reviewer_9

1. If k is unknown, but a bound k_max is known, our algorithm returns a mixture with at most k_max components, but may not be able to find the right k. However, no algorithm can find the right k, since some components may have zero or close to zero weight.

2. We are working on extending the results to non-spherical Gaussians. But as in most problems on Gaussian mixtures, we believe that studying spherical Gaussians is the first step.

3. Yes, for fixed d, the value of error delta would be at most 1/3. But For smaller values of delta, we run the algorithm with error 1/3 and repeat it O(log 1/delta ) times to choose a set of candidate mixtures F_{delta}. By the Chernoff-bound with error delta, F_{delta} contains a mixture epsilon-close to the underlying mixture. Finally, we run MODIFIED SCHEFFE on F_{delta} to obtain a mixture that is close to the underlying mixture. By the union bound and Lemma 1, the error of the new algorithm is <= 2 delta.